# Rumicidins are a family of mammalian host-defense peptides plugging the 70S ribosome exit tunnel

Pavel V. Panteleev [1,8], Eugene B. Pichkur[2,8], Roman N. Kruglikov[1], Alena Paleskava [2], Olga V. Shulenina[2], Ilia A. Bolosov[1], Ivan V. Bogdanov [1], Victoria N. Safronova[1], Sergey V. Balandin [1], Valeriya I. Marina[3], Tatiana I. Kombarova[4], Olga V. Korobova[4], Olga V. Shamova [5], Alexander G. Myasnikov[2], Alexander I. Borzilov[4], Ilya A. Osterman[3,6], Petr V. Sergiev [3,6], Alexey A. Bogdanov[1,3], Olga A. Dontsova[1,3,6], Andrey L. Konevega [2] ✉ & Tatiana V. Ovchinnikova [1,3,7] ✉

The antimicrobial resistance crisis along with challenges of antimicrobial discovery revealed the vital necessity to develop new antibiotics. Many of the animal proline-rich antimicrobial peptides (PrAMPs) inhibit the process of bacterial translation. Genome projects allowed to identify immune-related genes encoding animal host defense peptides. Here, using genome mining approach, we discovered a family of proline-rich cathelicidins, named rumicidins. The genes encoding these peptides are widespread among ruminant mammals. Biochemical studies indicated that rumicidins effectively inhibited the elongation stage of bacterial translation. The cryo-EM structure of the *Escherichia coli* 70S ribosome in complex with one of the representatives of the family revealed that the binding site of rumicidins span the ribosomal A-site cleft and the nascent peptide exit tunnel interacting with its constriction point by the conservative Trp23-Phe24 dyad. Bacterial resistance to rumicidins is mediated by knockout of the SbmA transporter or modification of the MacAB-TolC efflux pump. A wide spectrum of antibacterial activity, a high efficacy in the animal infection model, and lack of adverse effects towards human cells in vitro make rumicidins promising molecular scaffolds for development of ribosome-targeting antibiotics.

The rapid growth of antibiotic resistance among bacterial pathogens, along with the challenges of novel antimicrobial discovery, requires development of new anti-infective agents[1]. About half of conventional antibiotics, most of which are of natural origin, target the bacterial ribosome. An essential role of ribosomes in protein biosynthesis and a low mutational propensity still make them an ideal target for antimicrobials[2]. It has been shown that many genetically encoded endogenous animal host-defense peptides are enriched with proline

[1]M.M. Shemyakin & Yu.A. Ovchinnikov Institute of Bioorganic Chemistry, the Russian Academy of Sciences, Moscow, Russia. [2]Petersburg Nuclear Physics Institute named by B.P. Konstantinov of NRC "Kurchatov Institute", Gatchina, Russia. [3]Lomonosov Moscow State University, Moscow, Russia. [4]State Research Center for Applied Microbiology & Biotechnology (SRCAMB), Obolensk, Russia. [5]Institute of Experimental Medicine, Saint Petersburg, Russia. [6]Center of Life Sciences, Skolkovo Institute of Science and Technology, Skolkovo, Russia. [7]Department of Biotechnology, I.M. Sechenov First Moscow State Medical University (Sechenov University), Moscow, Russia. [8]These authors contributed equally: Pavel V. Panteleev, Eugene B. Pichkur. ✉e-mail: konevega_al@pnpi.nrcki.ru; ovch@ibch.ru

and arginine residues and inhibit the translation process. Ribosome-targeting proline-rich antimicrobial peptides (PrAMPs) from animals can be divided into two groups based on their origin: oncocin- and apidaecin-like peptides from insects, and cathelicidins from Cetartiodactyla mammals[3]. Cathelicidins are synthesized as prepropeptides containing an N-terminal signal peptide, a conservative cathelin-like domain (CLD), and a C-terminal mature AMP, which is enzymatically cleaved off once the peptide is secreted[4].

To date, two main mechanisms of translation inhibition have been described for PrAMPs: class I peptides (oncocins, Bac7-like cathelicidins) inhibit the elongation phase by sterically preventing the accommodation of the first incoming aminoacyl-tRNA into the ribosomal A site[5], whereas class II peptides (apidaecins, drosocin) prevent dissociation of the release factors during the termination phase of translation[6,7]. In general, PrAMPs are characterized by a capacity to form multiple interactions within the nascent peptide exit tunnel (NPET) that significantly reduces the risk of bacterial resistance due to single modifications or spontaneous mutations of the nucleotides in the 23S rRNA.

Search for natural PrAMPs is an important step toward the design of highly effective translation inhibitors. To develop PrAMP-based antibiotics by medicinal chemistry approaches, a battery of molecular scaffolds needs to be created and optimized for the purpose of enhancing their efficiency and getting the best result. Intensive development of the genome projects worldwide[8,9] allows fast identification of immune-related genes encoding animal host-defense peptides. In particular, genome mining reveals a significant potential for the discovery of proline-rich cathelicidins due to the high conserved structure of the cathelin-like domain and quite simple organization of the corresponding gene clusters.

Here, we describe a family of relatively small proline-rich cathelicidins widespread among ruminant species. These peptides, named rumicidins, inhibit the formation of the first peptide bond presumably due to interference with the accommodation of aminoacyl-tRNA into the A site of the ribosome. Our cryo-EM data reveals that rumicidins bind to the bacterial ribosome in the NPET similar to other known PrAMPs but form unique interactions with the elements of the 50S ribosomal subunit. The obtained data on the biological activity of rumicidins allows us to consider them as a quite safe and effective ribosome-targeting antimicrobials.

## Results

### Genes encoding for a family of proline-rich cathelicidins are widespread among ruminants

We used the TBLASTN program to identify cathelicidin genes in the whole-genome sequencing (WGS) database using the conservative cathelin-like domain (CLD) fragment FTVKETVCPRTSPQPPEQCDFKE encoded by the nucleotide sequence located in the second exon of the cattle procathelicidin-3 (the Bac7 precursor). We used this sequence as a query against all Cetartiodactyla WGS projects deposited in NCBI. The hit DNA contigs were analyzed and the obtained database of translated proline-rich cathelicidins was classified into structural families[10]: Bac4-, Bac5-, Bac6-, Bac7-, P9-, ChBac3.4-, prophenin-like peptides, and several additional ones. A widespread family of relatively short AMPs (28-30 residues long) was chosen for further study. The discovered peptides are similar in length to cetacean Bac7-like peptides, but lack the conserved consensus fragment (R/K)XX(R/Y)LPRPR required for strong binding of class I PrAMPs in the NPET[11,12]. We reasoned that the peptides belonging to this family might have unique binding site(s) and a distinct mechanism of action.

In this study, 147 Cetartiodactyla species were analyzed, and 65 of them were found to have these cathelicidin genes (Fig. 1). The genes were found only in ruminants: in representatives of almost all sub-families of bovids, as well as in musk deers, pronghorn, and cervids. Previously, in refs. 13,14, paralogous pseudogenes named *CATHL3L2*

and *CATH8* were described in the *Bos taurus* genome, which encode preprocathelicidins belonging to this family. Therefore, the discovered ruminant genes we designated as *CATHL(3L2/8)*. Among the found sequences, the major part belonged to pseudogenes (ψ), and only a quarter of the species (18 of 65) had presumably intact genes which displayed all characteristics of functional cathelicidin genes (Supplementary Fig. 1, **Supplementary Discussion**). Recently, the expression of one of *CATHL(3L2/8)* genes has been shown in the musk deer *Moschus berezovskii*[15]. We also performed de novo assembly of transcriptomes of several ruminant species and found target sequences (Supplementary Table 1). Analysis of ψ*CATHL(3L2/8)* pseudogenes revealed different variants of gene inactivation, including nonsense mutations (premature termination codon, indel) as well as start codon mutations (Supplementary Fig. 2 and Supplementary Data 1).

The family of PrAMPs encoded by *CATHL(3L2/8)*-like genes was named as rumicidins (**rumi**nant cathel**icidins**). In this study, we investigated three peptides: the rumicidin family consensus peptide identical to the peptide from the Tibetan antelope chiru *Pantholops hodgsonii* named rumicidin-1 and the most structurally distant orthologs identified in the genomes of the African antelope hirola *Beatragus hunteri* and the American pronghorn *Antilocapra americana* and named as rumicidin-2 and rumicidin-3, respectively (Fig. 2a). Well-studied proline-rich cathelicidins mini-ChBac7.5α, Bac7[1-22], and PR-39[1-22] were chosen as the reference class I PrAMPs (Fig. 2a). These truncated variants were shown to retain antibacterial activity of the wild-type peptides[16,17]. PrAMPs used in this study were produced in *E. coli* BL21 (DE3) cells using lactose-based autoinduction medium[18]. To facilitate the purification process and improve the final yield, the recombinant peptides were obtained as fusion proteins with the N-terminal His-tag and thioredoxin A. After cleavage of the fusion proteins with cyanogen bromide, the corresponding mature PrAMPs were purified by reversed-phase high-performance liquid chromatography (RP-HPLC) to reach a purity of ≥98%. Final yields of the recombinant cathelicidins and their analogs ranged from 3 to 12 mg per 1 L of the culture medium. The obtained peptides were analyzed by Tricine-SDS-PAGE (Supplementary Fig. 3) and MALDI-TOF mass-spectrometry (Supplementary Table 2).

### Rumicidins effectively inhibit the elongation stage of protein synthesis in bacteria

Taking into account the previously reported data on binding of the PrAMPs to the 70S ribosome, we first tested the ability of rumicidins to inhibit protein biosynthesis in vitro (Fig. 2b). The experiment was carried out using *E. coli* BL21(DE3) Star cell-free protein synthesis system (CFPS, coupled transcription/translation system) expressing the enhanced green fluorescent protein (EGFP). Interestingly, high structure variability of the N-termini of the studied peptides did not significantly alter the CFPS inhibition profile. We also tested the ability of rumicidins to suppress translation of the luciferase mRNA in vitro. It was shown that all the peptides almost fully and rapidly inhibited the process at the concentration of 5 μM (Fig. 2c). To verify the proposed mode of action of rumicidins in live bacteria, we tested their activity using an *E. coli*-based double-reporter system (Fig. 2d). As expected, rumicidins like Bac7[1-22] and erythromycin strongly induced biosynthesis of Katushka2S, but not of RFP, which indicated that these peptides inhibited protein synthesis in vivo as well.

To identify the step of translation specifically inhibited by rumicidins, we used a primer-extension inhibition (toe-printing) assay. The addition of rumicidins as well as Bac7[1-22] at a concentration of 5 μM (5×MIC, minimum inhibitory concentration) to a PURExpress cell-free transcription-translation system programmed with the ermCL mRNA[19] resulted in the ribosome stalling at the AUG start codon (Fig. 2e). Similar patterns were shown by toe-printing with RST1 mRNA[20] (Supplementary Fig. 4). To check an ability of rumicidins to influence formation of the first peptide bond, we compared the production of

fMet-[$^{14}$C]Val dipeptide on intact or inhibited with rumicidin-1 or amicoumacin A ribosomal complexes (Supplementary Fig. 5). Antibiotic amicoumacin A is known for its multifaceted action on the ribosome

with major effect on translocation ensuring unaltered reactions of A-site binding and peptide bond formation[21]. The addition of 1 μM rumicidin-1 to the functional ribosomal complex severely impaired the

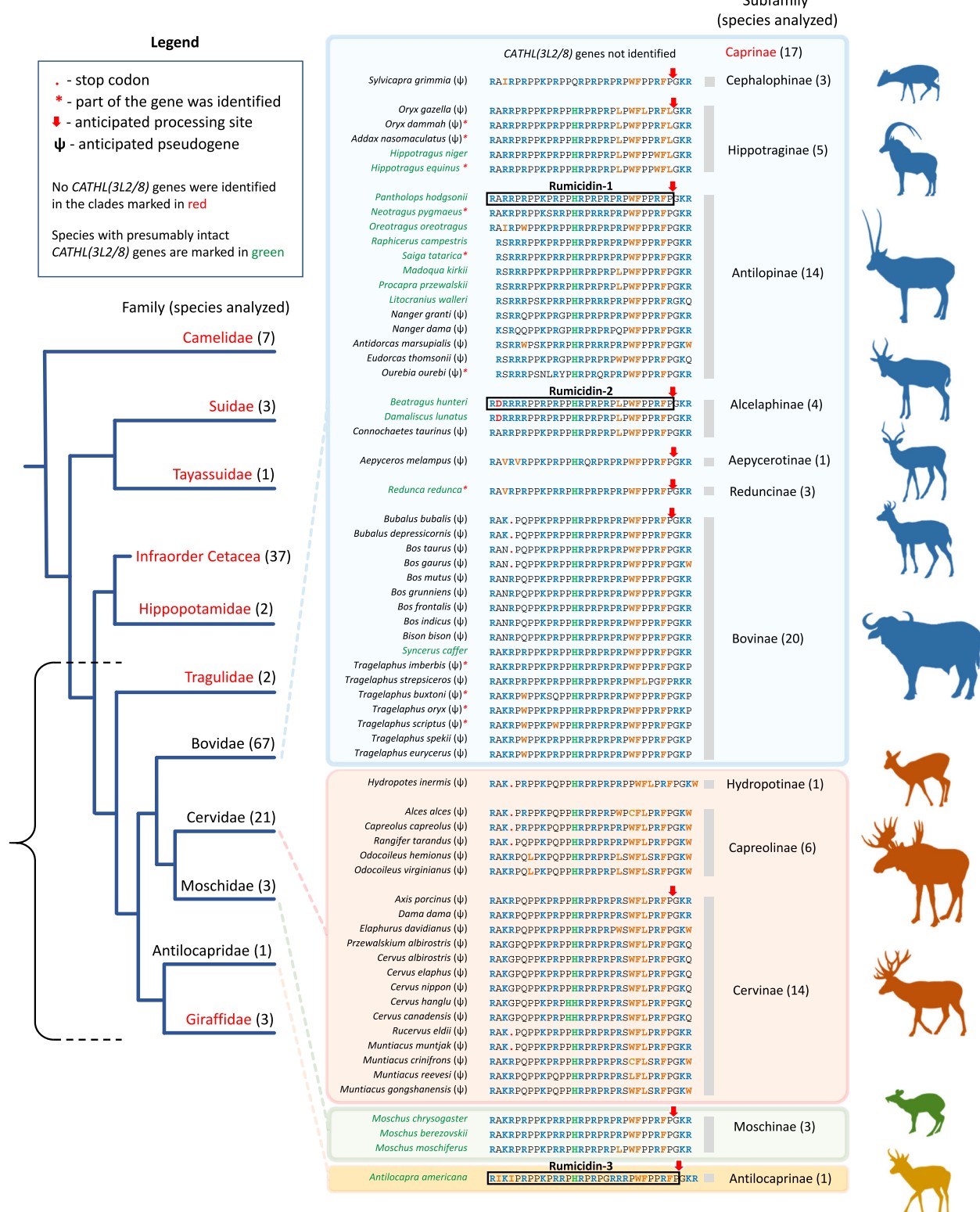

**Fig. 1 | Analysis of a family of proline-rich cathelicidin orthologs identified in ruminants.** Amino acid sequences of rumicidins after in silico processing of corresponding precursor proteins by elastase are presented in the colored boxes. The description box is presented at the top. Most of the rumicidins bear the *C*-terminal GKR tripeptide, which is a signal for an additional stage of post-translational modification by peptidylglycine alpha-amidating monooxygenase. This enzyme cleaves the *C*-terminal GKR (marked with a red arrow) and then amidates of the carboxyl group of the preceding amino acid residue. Such modifications occur in other proline-rich cathelicidins (for example, in the bovine Bac5 and porcine PR-39[10]) and may play an important role in providing resistance to carboxypeptidases.

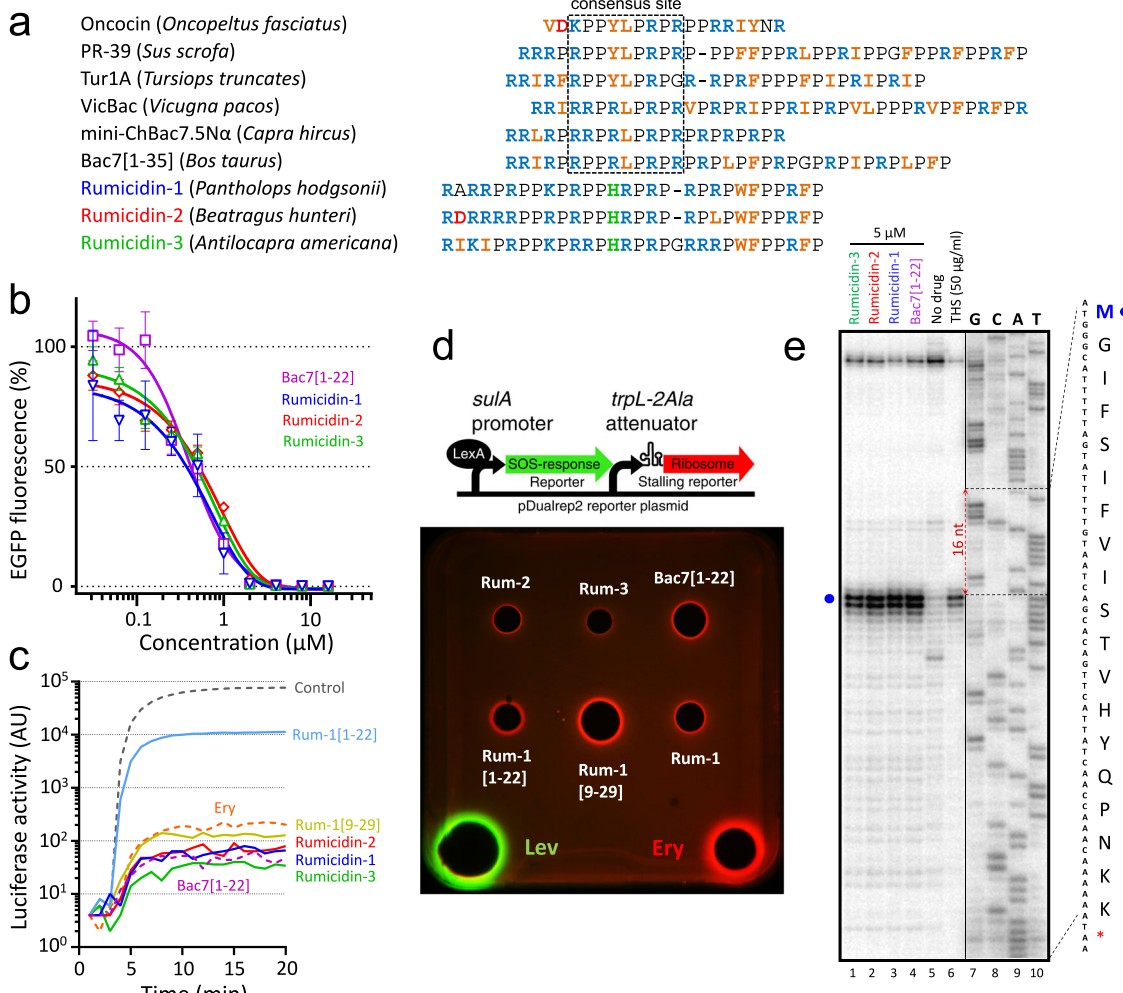

**Fig. 2 | Rumicidins inhibit protein synthesis in bacteria. a** Amino acid sequence alignment of the class I (oncocin/Bac7-like) PrAMPs. The consensus sequence essential for the inhibition of the protein biosynthesis by class I PrAMPs is marked with black dashed box. **b** Effects of rumicidins and Bac7[1-22] at different concentrations on the fluorescence resulting from the in vitro coupled transcription/translation of EGFP with the use of the *E. coli* BL21 (DE3) Star cell extract. At least two independent experiments were performed in triplicate, with the plotted points representing the mean value ± SD. **c** Kinetic curves showing inhibition the translation of the luciferase mRNA in vitro by erythromycin (Ery, 50 μg/ml), Bac7[1-22], rumicidins -1, -2, -3, and rumicidin-1 analogs 1-22 and 9-29 using an *E. coli* S30 extract. All the peptides were used at the concentration of 5 μM. AU - arbitrary units. **d** The induction of a two-color dual-reporter system sensitive to inhibitors of ribosome progression or of DNA replication. Induction of the expression of

Katushka2S (red color) is triggered by translation inhibitors, whereas RFP expression (green pseudocolor) is induced upon DNA damage. Control antibiotics levofloxacin (Lev) and erythromycin (Ery), and tested PrAMPs were placed on agar plate containing *E. coli* BW25113 Δ*tolC* cells transformed with the pDualrep2 plasmid[57]. **e** Ribosome stalling by rumicidins on ermCL mRNA, as revealed by the reverse-transcription inhibition (toe-printing) assay in a cell-free translation system. Thiostrepton (THS) and Bac7[1-22] were used as control compounds inhibiting the elongation step of bacterial translation. Both of them allow ribosomes to initiate translation at the AUG start codon but prevent further progression of the ribosome either due to the blocked access of elongation factors by THS[76] or due to the inability of aminoacyl-tRNA to accommodate into the A site by Bac7[5]. Two independent experiments were performed, and the observed effects were the same. Source data are provided as a Source Data file.

reaction resulting in essentially no dipeptide formed (7 ± 1%). At the same time, amicoumacin A containing ribosome complexes were as effective in the reaction of peptide bond formation as intact complexes with the amount of dipeptide formed equal to 72 ± 7% and 77 ± 5%, respectively. Taken together, our results suggest that rumicidins display their antibacterial activity via inhibition of protein biosynthesis. In particular, they block the first peptide bond formation and arrest elongation similar to other known class I PrAMPs.

## The *N*-terminal regions of rumicidins are not essential for translation inhibition

To verify roles of different structural elements of rumicidins both in translation inhibition and in antibacterial action, we performed a structure-activity relationship (SAR) study of rumicidin-1 (Fig. 3). Surprisingly, the IC$_{50}$ values of the truncated analogs [residues 1-16] and

[residues 1-22] are 100- and 10-fold higher, respectively, compared to the full-length rumicidin-1 [residues 1-29], in distinction from Bac7, for which the first 16 residues retain a high efficiency of translation inhibition[11,22]. Interestingly, the presence of the terminal extra (PR)$_3$ fragment provides a significant inhibition effect of the analog 1-22 (IC$_{50}$ of 7.2 ± 0.3 μM) as compared with that of the analog 1-16 (IC$_{50}$ of 44.6 ± 9.8 μM). Previous studies revealed that poly-PR peptides strongly bound to the polypeptide tunnels of both 70S and 80S ribosomes[23].

Notably, the deletion of four *N*-terminal residues (RRIR) in the peptides Bac7[5-23] and Bac7[5-35] greatly diminished their antimicrobial activity against *E. coli* strains (≥16-fold increase in MIC), albeit a sufficient length was the case[17]. This highly cationic fragment is important for both high affinity binding of Bac7 to the ribosome and its penetration inside bacterial cells[5]. The effect of *N*-terminal shortening

Bac7[1-22]  RRIRPRPPRLPRPRPRPLPFPR
consensus site

| Sequence | Rumicidin-1 variant | Protein synthesis inhibition (µM)* | Minimum inhibitory concentration (µM)** | |
|---|---|---|---|---|
| | | | MHB | MHB+ 0.9% NaCl |
| RARRPRPPKPRPPHRPRPRPRPWFPPRFP | WT | 2 | 2 | 4-8 |
| RARRPRPPKPRPPRLPRPRPRPWFPPRFP | [H14R,R15L] | 2 | 2 | 4-8 |
| RARRPRPPKPRPPARPRPRPRPWFPPRFP | [H14A] | 4 | 2 | 8 |
| RARRPRPPKPRPPHRPRPRPRAWFPPRFP | [P22A] | 4 | 4 | 8-16 |
| RARRPRPPKPRPPHRPRPRPRPAFPPRFP | [W23A] | 4 | 2 | 16-32 |
| RARRPRPPKPRPPHRPRPRPRPAAPPRFP | [W23A,F24A] | 8 | 4 | 32 |
| RARRPRPPKPRPPHRP | [1-16] | 256 | 256 | >256 |
| RARRPRPPKPRPPHRPRPRPRP | [1-22] | 32 | 16 | 256 |
| RPRPPKPRPPHRPRPRPRPWFPPRFP | [4-29] | 4 | 4 | 16 |
| RPPKPRPPHRPRPRPRPWFPPRFP | [6-29] | 4 | 4 | 16-32 |
| KPRPPHRPRPRPRPWFPPRFP | [9-29] | 4 | 8 | 32 |
| KPRPPHRPRPRPRPAAPPRFP | [9-29, W23A,F24A] | 32 | 8 | >256 |
| RPPHRPRPRPRPWFPPRFP | [11-29] | 32 | 8 | 64 |
| PHRPRPRPRPWFPPRFP | [13-29] | 64 | 32 | >256 |
| RPRPRPRPWFPPRFP | [15-29] | 256 | 32 | >256 |

**Fig. 3 | Structure-activity relationship study of rumicidin-1.** * - a minimum concentration that provides >90% inhibition of the EGFP synthesis in the CFPS system. ** - MICs were determined against *E. coli* BW25113 in the Mueller-Hinton broth (MHB) without or supplemented with 0.9% NaCl.

of rumicidin-1 was not so marked: activities of the analogs 4-29 and 6-29 decreased only two-fold, while the MIC values of the analogs 9-29 and 11-29 against *E. coli* were ≥4-fold higher than those of the wild-type rumicidin-1. While both terminal analogs 1-22 and 9-29 were able to inhibit translation that was shown both in vitro (Fig. 2c; Supplementary Fig. 6a) and in vivo (Fig. 2d), the latter peptide was as effective as the full-length rumicidin-1. This points to key role of contacts between the ribosome and *C*-terminal residues of rumicidin-1 but not with *N*-terminal ones. As expected, the addition of the rumicidin-1 analog 9-29 led to a strong AUG toe-print signal comparable to that of the wild-type peptide, whereas a weak band was observed for the *N*-terminal peptide analog 1-22 (Supplementary Fig. 4). Thus, the *N*-terminus is likely needed for cellular uptake rather than for the ribosome binding, with the length of the terminus correlating with an antibacterial activity. Interestingly, the shortest *C*-terminal peptide analogs 13-29 and 15-29 were almost unable to inhibit both bacterial translation and cell growth, which also indicated an important role of amino acid residues in the central part of the peptide for interaction with the ribosome. However, the findings left open the question regarding the orientation of rumicidins in the NPET.

## Rumicidins combine the binding mode of several known PrAMPs

To determine the mode of binding of rumicidins to the *E. coli* 70S ribosome, we obtained a cryo-EM structure of rumicidin-2 bound to the functional mRNA-programmed ribosomal complex with fMet-tRNA$^{fMet}$ in the P site. This peptide has a unique *N*-terminal part and the lowest homology with known PrAMPs among all representatives of this structural family. Focused refinement of the 50S subunit yielded a 1.95 Å cryo-EM density map (Supplementary Fig. 7, Supplementary Table 3). As expected from structural studies of other PrAMPs, rumicidin-2 binds to the 70S ribosome within the exit tunnel of the 50S, which serves as the path for a nascent chain (Fig. 4a). A distinct density observed within the ribosomal exit tunnel could be unambiguously assigned to residues 12-27 of rumicidin-2 bound in an extended conformation. Rumicidin-2 enters the exit tunnel in a reversed orientation relative to a nascent polypeptide chain (Fig. 4b) and utilizes multiple hydrogen bonding, stacking and van der Waals interactions to bottle it up (Fig. 4e, f). Our structure presents high-

resolution details for the core regions of the 50S and well-resolved parts of the rumicidin-2.

The most pronounced variability, including the number and composition of amino acid residues, was observed for the *N*-terminal parts of PrAMPs. The *N*-terminal part of rumicidin-2 consists of 11 amino acid residues, being the longest among all visualized PrAMPs. Although no density could be seen for this part of the peptide, presumably due to its mobility, according to structure similarity with other peptides we assume that rumicidin-2 reaches into the A-tRNA binding pocket (Fig. 4c, d, Supplementary Fig. 8).

The *C*-terminal parts of all previously described oncocin-like peptides do not appear to make any specific contacts with the ribosome. Surprisingly, we have found that rumicidin-2 has a clear density protruding into the depth of the exit tunnel interacting with its constriction point (Fig. 4f). The lower part of a spacer in a constriction site is formed by Trp23 and Phe24, which is a structural feature of all rumicidins. Backbone-carbonyl oxygen of Pro22 forms a hydrogen bond with the Arg61 of the ribosome protein uL4 while the aromatic ring of Trp23 forms stacking interaction with nucleotide A751 of helix H34 of 23S rRNA and, in addition, hydrogen bonds with the carbonyl oxygen of Lys90 of the ribosomal protein uL22 (Fig. 4f). Thus, our data suggest that the extended loop of uL22 not only stabilizes the position of H34 backbone, facilitating the binding of peptide and 23S rRNA, but also interacts with rumicidin-2 directly. Phe24 stacks upon Thr65 of the ribosomal protein uL4 via C-H···π-interactions, narrowing down the constriction of the exit tunnel together with the Trp23 (Fig. 4f). Interestingly, insect-derived class II PrAMP, Api137 has a similar binding pattern despite the opposite orientation of the peptide in the tunnel[6]: Tyr7 stacks upon nucleotide A751 of H34 of 23S rRNA, while Arg10 forms a hydrogen bond with the residue Arg61 of the ribosomal protein uL4. At the end of the *C*-terminal part, carbonyl oxygen of Pro25 forms hydrogen bonds with Arg67 (uL4), which rotates its side-chain towards the backbone of rumicidin-2, although two alternative conformations of the Arg67 were observed in the structure.

Due to high sequence resemblance with other oncocin-like peptides, the central part of rumicidin-2 (residues 12-22) almost perfectly aligns with the residues 7-17 in Bac7[1-19][2,5] and thus makes similar contacts with the rRNA (Fig. 4d). His14 intercalates into the A-site cleft (Fig. 4e), a hydrophobic pocket formed by the nucleobases of A2451

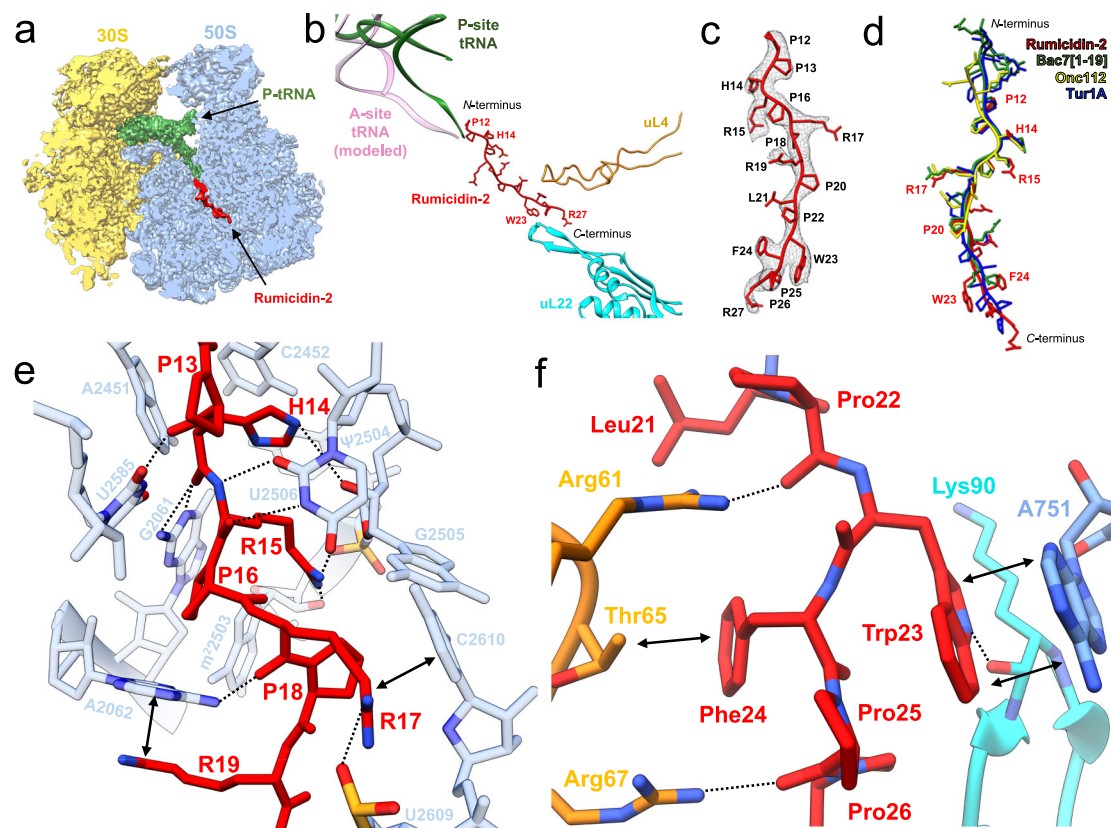

**Fig. 4 | Cryo-EM structure of rumicidin-2 and its interactions with the bacterial ribosome. a** Cross-section of the 70S initiation complex (50S subunit in blue, 30S subunit in yellow, fMet-tRNA$^{fMet}$ in green) with the rumicidin-2 (red) bound in the exit tunnel (PDB ID 9D89, EMD-46632). **b** Binding site of rumicidin-2 relative to fMet-tRNA$^{fMet}$ (green), uL4 (orange) and uL22 (cyan) ribosomal proteins. **c** Coulomb potential map of rumicidin-2 (gray mesh) determined with cryo-EM and a corresponding atomic model (red) of the well-resolved P12-R27 fragment of rumicidin-2. **d** Structural comparison of rumicidin-2 (red, PDB ID: 9D89) with Bac7 (green, PDB ID: 5HAU), Onc112 (yellow, PDB ID: 4Z8C), and Tur1A (blue, PDB ID: 6FKR) in the ribosomal exit tunnel. Similarly to other oncocin-like AMPs, rumicidins bind to the ribosomal peptide exit tunnel in the extended conformation and are oriented with the *N*-terminal region towards PTC. **e** Interactions of the central part of rumicidin-2 (captioned in red) with 23S rRNA (nucleotides are captioned in light blue, stacking interactions are shown as black arrows, hydrogen bonds are shown as dotted lines). **f** Interactions of rumicidin-2 (captioned in red) with uL4 protein (residues Arg61, Arg67 captioned in orange), uL22 (residues Lys90 captioned in cyan) and 23S rRNA (A751 captioned in blue).

and C2452, and occupies similar position to Arg9 in Bac7 and Tyr6 in Onc112[24] while forming H-bond with the O2′ atom of the ribose G2505. Side-chain of Arg15 forms a hydrogen bond with the backbone phosphate of G2505 closely resembling a position of the Leu7 in Onc112. Multiple bonds are formed between backbone amino/carbonyl groups and 23rRNA. Pro13 may form a hydrogen bond with U2585, although the latter is not sufficiently resolved. Backbone-carbonyl oxygen of Pro14 forms a hydrogen bond with G2061, while both backbone oxygen and nitrogen of Arg15 form hydrogen bonds with U2506. In the upper tunnel, Arg17 stacks upon C2610, reaching the backbone phosphate of U2609 identically to the Arg12 of Bac7 and Arg9 of Onc112, while backbone carbonyl oxygen forms a hydrogen bond with A2062. Clear density is observed for the side-chain of Arg19 and A2062 which interact by π-stacking identically to Arg16 of Bac7, implying an additional stabilization of the A2062 by this interaction. Contrary to other relatively well-defined arginine residues in the central part, we have found no density in the map for the side-chain of Leu21. This lack of coordination may arise as Leu and Arg are used interchangeably at this location in rumicidins (Supplementary Fig. 1).

## Mechanisms of bacterial resistance to rumicidin-1

The selection of resistant strains makes it possible to shed light both on the key targets of the antibiotic and on the mechanisms of protection against it in bacteria. At the first stage, we set a goal to obtain bacteria resistant to rumicidin-1 using the *E. coli* SQ110LPTD strain

which has compromised outer membrane and lacks 6 of 7 chromosomal *rrn* alleles encoding for rRNAs[20]. As a result, the final MIC value did not differ more than two-fold from the initial even after 20 passages in the Mueller-Hinton broth (MHB) supplemented with rumicidin-1.

Next, we applied *E. coli* 1057 strain[22] when performing resistance induction experiments (Supplementary Fig. 9a). This strain carried two well-characterized mutations in *gyrA* (S83L and D87N, Supplementary Fig. 10) causing a high-level fluoroquinolone resistance as well as a higher-than-normal spontaneous mutation rate[25]. Notably, ≥8-fold increase in the MIC value was registered after several passages in the medium containing 0.9% NaCl subjected to selection by PrAMPs. In this work, we identified *sbmA* frameshift (Δ2 bp, the strain R1) in the case of rumicidin-1 (Supplementary Table 4, Fig. 5a). Earlier, we found the V102E substitution in SbmA of the *E. coli* 1057 strain after treatment by caprine mini-ChBac7.5Nα[22]. SbmA is a homodimer proton-driven transporter found amongst some classes of Proteobacteria, which is involved in the uptake of non-ribosomal peptide antibiotics, bacteriocins, and PrAMPs[26,27]. Obviously, this transporter is under strong selective pressure when salt-containing media is used.

Our previous studies showed that in the absence of the salt in the medium PrAMPs were able to effectively act against strains with mutant SbmA[22,28]. Here, we found that rumicidin-1 also had the same MICs of 1 μM against both BW25113 and its Δ*sbmA* variant in salt-free MHB. Indeed, the absence of 0.9% NaCl in the media greatly affects

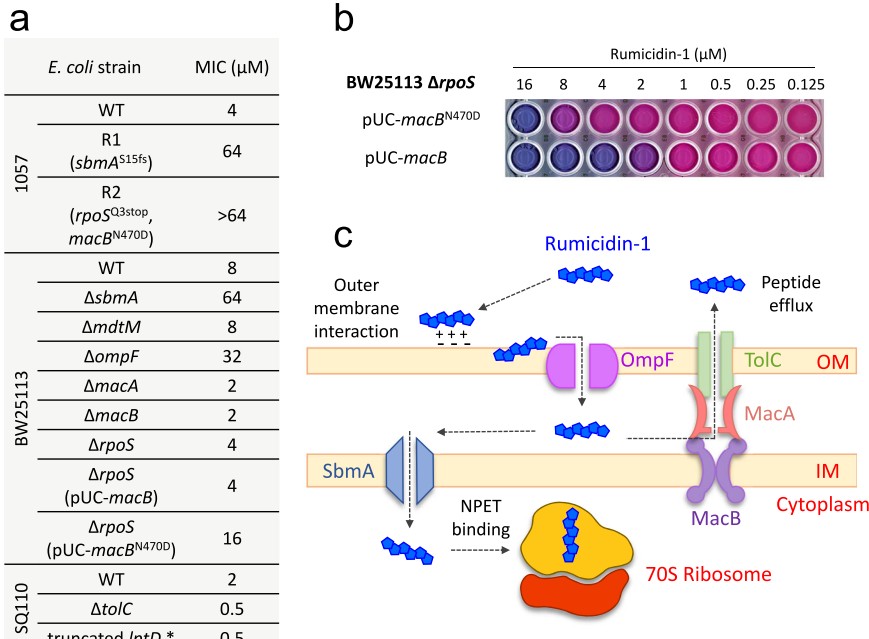

**Fig. 5 | Mechanisms of bacterial resistance development to rumicidins. a** MICs of rumicidin-1 against obtained rumicidin-1-resistant *E. coli* strains (R1 and R2) and other strains in the Mueller-Hinton broth supplemented with 0.9% NaCl. * - the *lptD* gene bears the deletion of 23 codons (Asp330-Asp352) which makes the outer membrane of the strain more permeable to antibiotics[20]. **b** Resazurin visualization of the antimicrobial microdilution test performed for rumicidin-1 against BW25113 Δ*rpoS* constructively expressing the additional allele of either *macB* or the mutant *macB*[N470D]. Viable cells reduce blue resazurin to pink resorufin. **c** The proposed mechanism of action of rumicidins against *E. coli*.

both the dynamics and main molecular targets of selective pressure under PrAMPs treatment (Supplementary Fig. 9a). We found no changes in MIC for mini-ChBac7.5Nα and 16-fold increase in MIC for rumicidin-1 (the strain R2). The whole-genome sequencing of the strain R2 (Fig. 5a) showed the point mutation in the *macB* gene [N470D] and the nonsense point mutation in the *rpoS* gene [Q3stop] (Supplementary Table 4). RpoS is a stress sigma factor that controls the expression of many genes implicated in the survival of the cell under suboptimum or stressful growth conditions[29]. Here, we found that *E. coli* BW25113 Δ*rpoS* was even 2-fold more susceptible to PrAMPs as compared to the wild-type strain. MacB is an ABC transporter that collaborates with the MacA adapter protein and the TolC exit duct and, in this way, provides an efflux of antibiotics out of the bacterial cell[30]. The MacB transmembrane domain lacks a central cavity through which antibiotics could be passed inside the cell but conveys conformational changes upon ATP binding[30]. Indeed, the gene knockout of either *macA* or *macB* resulted in a 4-fold decrease in MIC for rumicidin-1. Perhaps, MacAB-TolC can efflux PrAMPs from periplasmic space, while the N470D substitution in the MacB seems to increase the selectivity of the transporter to AMPs (Supplementary Fig. 11 **and Supplementary Discussion**). To test this hypothesis, we checked the activity of the BW25113 Δ*rpoS* strains having additional plasmid-borne allele encoding either the wild-type MacB or the mutant MacB[N470D]. The target genes were expressed using three variants of complementation plasmids: under the strong constitutive artificial promoter J23119[31], or the inducible leaky hybrid T5lac promoter, or the tightly regulated arabinose promoter (Supplementary Fig. 12). In all three cases, we found a 4-fold increase in MIC for the strain expressing mutant variant of the MacB domain as compared with the wild-type (the observed increase in MIC from 4 to 16 µM for J23119 is presented in Fig. 5a,b).

## Rumicidins have a wide spectrum of antibacterial activity
To estimate the therapeutic potential of rumicidins as candidate compounds for combating bacterial infections, we assessed their antimicrobial activity against a wide panel of strains. The resulting MIC heatmap is shown in Fig. 6a. As expected, antibacterial activities of rumicidins, like those of known PrAMPs, decreased in most cases in the presence of salt, which might inhibit the adsorption of the peptides to the bacterial surface. Besides, their activities against Gram-negative bacteria deficient in SbmA protein (*Pseudomonas aeruginosa* and *Proteus mirabilis*) were also less pronounced. Among all tested PrAMPs, rumicidin-3 from *A. americana* was of particular practical interest, having a minimal median MIC of about 2 µM (Fig. 6a). Notably, rumicidins also exhibited significant activities against Gram-positive bacteria, in particular micrococci, bacilli, and mycobacteria. All rumicidins have a similar activity against *Mycobacterium phlei* with MICs of 0.25-1 µM in the salt-free MHB and of 4-8 µM in the medium supplemented with 0.9% NaCl. We also accessed activities of the peptides against another fast-growing mycobacterial strain *M. smegmatis* mc(2) 155 and found rumicidin-1 to be the most active variant with MICs of 0.125-0.5 µM depending on the test medium (Supplementary Table 5). Activities of two other rumicidins were at least 4-fold lower. As PR-39 is the only described PrAMP having an activity against mycobacteria (MIC ~ 10 µM[32]), we used its analog PR-39[1-22] as a reference peptide. Notably, similar MICs of 8–16 µM were determined when testing activities against *M. smegmatis* (Supplementary Table 5). In total, these preliminary data make rumicidin-1 a comparatively effective and promising antimycobacterial agent as well.

Considering the presence of aromatic Trp and Phe residues in the structure of all rumicidins, we investigated the ability of these peptides to damage *E. coli* membranes. The effect of rumicidin-1 on the cytoplasmic membrane was quite modest (less than 20% of permeabilized cells) even at a concentration of 64 µM, which is 16-fold higher than the MICs against *E. coli* (Supplementary Fig. 13). The obtained results agree with the previous data on several cetacean ribosome-targeting PrAMPs, which did not alter cytoplasmic membrane permeability at MICs[12]. Notably, we found no significant differences in the action of rumicidin-1 and Bac7[1-22] on both the outer and inner membrane of *E. coli* (Supplementary Fig. 14). Interestingly, both peptides at concentrations ≥ MICs were shown to be quite effective in damaging the

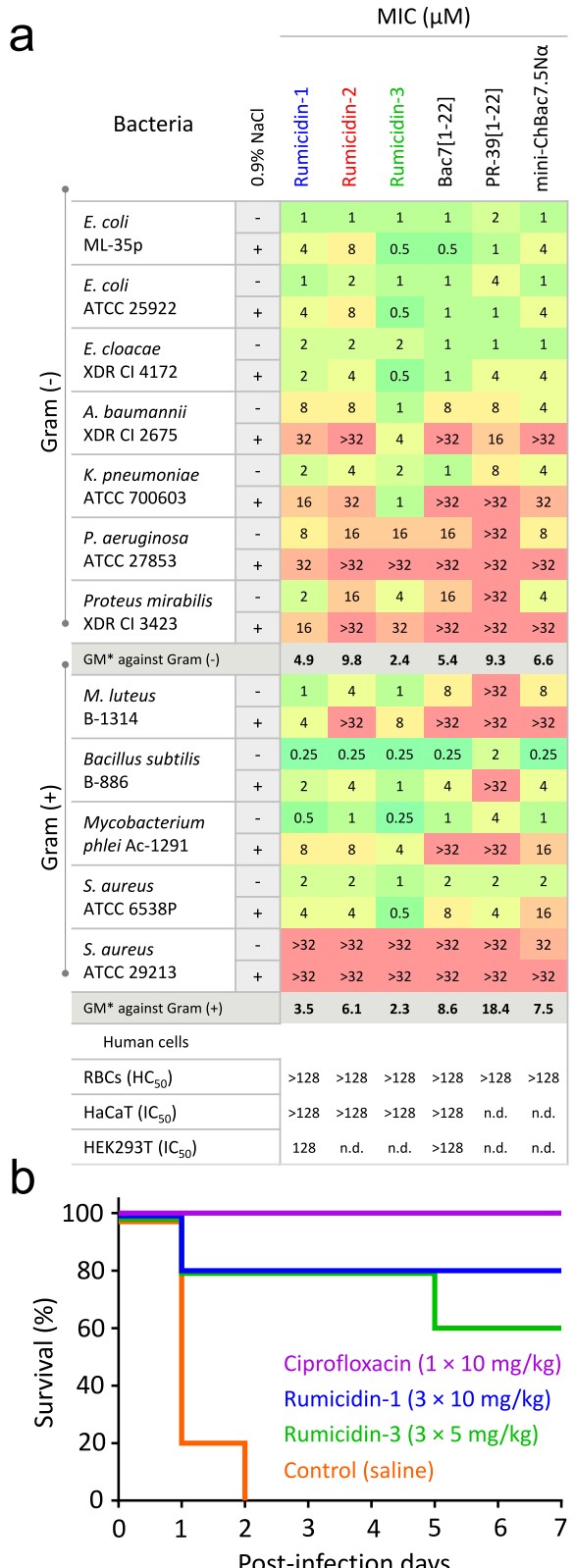

**a**

| | | MIC (μM) | | | | | | |
|---|---|---|---|---|---|---|---|---|
| Bacteria | 0.9% NaCl | Rumicidin-1 | Rumicidin-2 | Rumicidin-3 | Bac7[1-22] | PR-39[1-22] | mini-ChBac7.5Nα |
| *E. coli* ML-35p | - | 1 | 1 | 1 | 1 | 2 | 1 |
| | + | 4 | 8 | 0.5 | 0.5 | 1 | 4 |
| *E. coli* ATCC 25922 | - | 1 | 2 | 1 | 1 | 4 | 1 |
| | + | 4 | 8 | 0.5 | 1 | 1 | 4 |
| *E. cloacae* XDR CI 4172 | - | 2 | 2 | 2 | 1 | 1 | 1 |
| | + | 2 | 4 | 0.5 | 1 | 4 | 4 |
| *A. baumannii* XDR CI 2675 | - | 8 | 8 | 1 | 8 | 8 | 4 |
| | + | 32 | >32 | 4 | >32 | 16 | >32 |
| *K. pneumoniae* ATCC 700603 | - | 2 | 4 | 2 | 1 | 8 | 4 |
| | + | 16 | 32 | 1 | >32 | >32 | 32 |
| *P. aeruginosa* ATCC 27853 | - | 8 | 16 | 16 | 16 | >32 | 8 |
| | + | 32 | >32 | >32 | >32 | >32 | >32 |
| *Proteus mirabilis* XDR CI 3423 | - | 2 | 16 | 4 | 16 | >32 | 4 |
| | + | 16 | >32 | 32 | >32 | >32 | >32 |
| GM* against Gram (-) | | **4.9** | **9.8** | **2.4** | **5.4** | **9.3** | **6.6** |
| *M. luteus* B-1314 | - | 1 | 4 | 1 | 8 | >32 | 8 |
| | + | 4 | >32 | 8 | >32 | >32 | >32 |
| *Bacillus subtilis* B-886 | - | 0.25 | 0.25 | 0.25 | 0.25 | 2 | 0.25 |
| | + | 2 | 4 | 1 | 4 | >32 | 4 |
| *Mycobacterium phlei* Ac-1291 | - | 0.5 | 1 | 0.25 | 1 | 4 | 1 |
| | + | 8 | 8 | 4 | >32 | >32 | 16 |
| *S. aureus* ATCC 6538P | - | 2 | 2 | 1 | 2 | 2 | 2 |
| | + | 4 | 4 | 0.5 | 8 | 4 | 16 |
| *S. aureus* ATCC 29213 | - | >32 | >32 | >32 | >32 | >32 | 32 |
| | + | >32 | >32 | >32 | >32 | >32 | >32 |
| GM* against Gram (+) | | **3.5** | **6.1** | **2.3** | **8.6** | **18.4** | **7.5** |
| Human cells | | | | | | | |
| RBCs (HC$_{50}$) | | >128 | >128 | >128 | >128 | >128 | >128 |
| HaCaT (IC$_{50}$) | | >128 | >128 | >128 | >128 | n.d. | n.d. |
| HEK293T (IC$_{50}$) | | 128 | n.d. | n.d. | >128 | n.d. | n.d. |

Gram (-), Gram (+) labels on left side.

**b**

Survival (%) vs Post-infection days:
- Ciprofloxacin (1 × 10 mg/kg)
- Rumicidin-1 (3 × 10 mg/kg)
- Rumicidin-3 (3 × 5 mg/kg)
- Control (saline)

**Fig. 6 | A therapeutic potential of rumicidins. a** Antibacterial activity and cytotoxicity of rumicidins in comparison with known PrAMPs. MICs were determined in the rich Mueller-Hinton broth (MHB) ± 0.9% NaCl. *The geometric mean of MICs (GM) against tested strains (when the value > 32 μM was observed, the MIC of 64 μM was used for calculation). **b** Survival rates of BALB/c mice (*n* = 5) infected intraperitoneally (IP) with *E. coli* ATCC 25922 (10⁶ bacteria in the presence of 2.5% mucin). Ciprofloxacin (the single dose of 10 mg/kg) and saline were used as positive and negative control, respectively. Mice were controlled once daily for their health status during 7 post-infection days. A high CFU burden in mice from the negative control group was verified. The absence of *E. coli* in spleen after euthanizing of all survived animals was controlled.

### Cytotoxic and immunomodulatory activity of rumicidin-1

It is known that most PrAMPs have no pronounced toxicity toward mammalian cells[12]. A quite effective inhibition of eukaryotic ribosome translation by the truncated analogs of Bac7 was shown[5]. At the same time, the absence of toxicity against HaCaT and erythrocytes was demonstrated for the Bac7[1-35][12]. Therefore, the antibacterial selectivity of the peptide is achieved due to a low efficiency of penetration into mammalian cells rather than to a low affinity for 80S and mitochondrial ribosomes. In this study, we showed that rumicidins as well as the control peptide Bac7[1-22] lysed less than 2% of red blood cells (RBCs) at the concentration of 128 μM (Supplementary Fig. 15). Very modest cytotoxic effects of the peptides against HaCaT (IC$_{50}$ > 128 μM) and HEK293T (IC$_{50}$ ~ 128 μM) cell lines were observed as well (Fig. 6a). Membrane-active PrAMPs were shown to undergo significant conformational changes when transiting from bulk solution into membrane-mimicking environments[12]. In contrast, circular dichroism (CD) spectra of rumicidin-1 were similar in water and zwitterionic dodecylphosphocholine (DPC) micelles (Supplementary Fig. 16), possibly suggesting the absence of specific interactions with neutral mammalian membranes.

Cathelicidins are known to have different immunomodulatory effects on various types of cells[33]. Given the absence of proline-rich cathelicidins in humans and, therefore, the likelihood of unpredictable effects on the human immune system, next, we assessed the effect of rumicidin-1 at a potential therapeutic concentration of 2 μM in blood stream on the expression profile of key cytokines and chemokines by immune and epithelial cells (**Supplementary Discussion**, Supplementary Fig. 17, Supplementary Table 6). The peptide was shown to display both pro- and anti-inflammatory action similar to human cathelicidin LL-37. Expectedly, the most significant effects on human cells were specifically triggered by LL-37. However, we also found a number of pro-inflammatory effects unique to rumicidin-1. In particular, it is able to significantly induce production of the chemotactic IL-8/CXCL8 by the epithelial Caco-2 cells as well as to inhibit the soluble interleukin-1 receptor antagonist (IL-1RA) production by different cell lines. Together, our data indicate the lack of significant adverse immunomodulatory effects of rumicidin-1 towards human cells in vitro.

### Rumicidins show an efficacy in the mouse septicemia model

Next, we examined an efficacy of rumicidins in the mouse septicemia model (Fig. 6b). In comparison with membrane-active peptides, PrAMPs are known as low-toxic substances having maximum tolerated doses (MTD) of above 50 mg/kg[34,35]. Here, we used two peptides – rumicidin-1 and -3 and applied each of them three times within the first day after infection. Each injection of rumicidin-1 was administrated at the dosage of 10 mg per kg of body weight and of 5 mg/kg for rumicidin-3 as the most active variant. Intraperitoneal infection of BALB/c mice with *E. coli* ATCC 25922 in the presence of mucin resulted in the death of all 5 mice within two days if treated using the vehicle control (saline) and survival of all mice when treated with 10 mg/kg ciprofloxacin. We demonstrated a therapeutic efficacy of 80% and 60% when applying the triple dose of rumicidin-1 and rumicidin-3,

outer membrane of bacterial cells washed with sodium phosphate buffer, whereas the addition of salts at physiological concentrations, in particular, 0.9% NaCl and/or divalent cations (0.5 mM Ca$^{2+}$ and 0.5 mM Mg$^{2+}$) minimized the activity. Perhaps, translocation of rumicidin-1 into the periplasmic space depends on electrostatic interaction with the outer membrane followed by porin-mediated uptake (the *ompF* gene knockout resulted in a 4-fold increase in MIC, Fig. 5a).

respectively, after one-week experiment. Our results are comparable to those for Bac7PS - a lead candidate compound selected by high-throughput screening of Bac7 analogs[35]. These data suggest that rumicidins are promising natural compounds for developing antimicrobials.

## Discussion

Many animal genomes contain several distinct AMP gene families. As a result, host organisms deploy them in synergistic mixtures that minimize the probability of evolution of bacterial resistance in nature[36], and PrAMPs are the key ingredients in these cocktails in artiodactyls and insects[22,37]. Here, we discovered and investigated a family of ribosome-targeting PrAMPs named rumicidins. The high distribution of corresponding orthologous genes among ruminants suggest their appearance before the split of pronghorns and other pecorans (all ruminants excluding Tragulidae) at the Oligocene/Miocene boundary[38,39].

Our results obtained both in vitro (Fig. 2b, c) and in vivo (Fig. 2d) suggested that rumicidins exerted their antibacterial activity via inhibition of protein biosynthesis. In particular, they inhibit the first peptide bond formation and block the elongation stage of translation, similarly to other known oncocin- and Bac7-like PrAMPs. Notably, complete translational inhibition was achieved at concentrations equal to or even lower than the MIC values against different *E. coli* strains. We cannot fully exclude any secondary intracellular targets (e.g., chaperone DnaK[3]), but in total, the obtained data allowed us to consider the ribosome as the main molecular target of rumicidins, which was also confirmed by cryo-EM data (Fig. 4). The structure of rumicidins consists of two functional parts: its central region enriched with PR repeats (which are essential for a peptide binding within the NPET[23]) is crucial for translation inhibition, whereas the *N*-terminal residues are likely to be important for the uptake of the peptide and provide low MICs against *E. coli*. Notably, these findings are in contrast to features of the known Bac7- and Bac5-like cathelicidins where the *N*-terminal residues are essential for ribosome binding[5,40,41]. The density was observed for the *C*-terminal fragment [12-27] in the case of rumicidin-2 (Fig. 4c), rationalizing our SAR data (Fig. 3). This fragment appears to be consistent with the sequence of the pharmacophore essential for the on-target activity: while the analog 11-29 is still able to impede protein synthesis and stop cell growth at a relatively low concentrations (≥8 μM, Fig. 3, Supplementary Fig. 6a), a subsequent shortening of rumicidin-1 leads to a dramatic loss of activity and an increase in MICs.

Superpositioning of structures of rumicidin-2 and other PrAMPs bound to bacterial ribosomes provides insight into the consensus sequence (R/K)XX(R/Y)LPRPR of oncocin- and Bac7-like peptides (class I PrAMPs) localized in the A-site cleft and the upper tunnel. According to the alignment shown in Fig. 2a, the key functional site Arg/Tyr-Leu changed to His-Arg in the case of rumicidins. The incorporation of Arg-Leu instead of His-Arg at the key site as well as the point substitution [H14A] did not influence the activity of rumicidin-1. This is in sharp contrast with the short PrAMPs like oncocin or Bac7[1-16], where amino acid residues in conservative sites were not tolerable even to a single substitution[11,42]. Next, we assessed a role of the conservative dyad Trp23-Phe24 forming key stacking interactions in the constriction site. Surprisingly, the analog of rumicidin-1 with a single substitution [W23A] exhibits a similar activity against *E. coli* in the salt-free MHB and strongly inhibits protein biosynthesis in vitro. A further modification of the dyad by a double substitution [W23A, F24A] resulted to a slight activity decrease. A 4-8-fold rise in MICs of both analogs in the test medium with an increased ionic strength is likely resulting from a weaker ability to interact/penetrate cell membranes (Fig. 3, Supplementary Figs. 13,14).

At the same time, the effectiveness of inhibition of protein biosynthesis is reduced significantly for the analog [W23A, F24A] as compared to the wild-type peptide (Supplementary Fig. 6b), that

points at a dual function of this hydrophobic fragment. A more pronounced decrease in both activities was found for the shortened analog 9-29 with a similar [W23A, F24A] modification of the dyad. To destabilize the interaction of the dyad in the tunnel, the Pro22 residue was also substituted with alanine, however only a 2-fold increase in MIC was detected. We can assume that extensive interactions with the 50S ribosomal subunit make rumicidins highly tolerable to amino acid substitutions (even at the key sites according to our cryo-EM data), which opens the door to further rational design of new analogs.

We did not find any mutations in ribosomal genes in the obtained rumicidin-resistant *E. coli* strains, which confirmed the high efficiency of multisite binding of these extended peptides in the NPET. Moreover, no increase in MIC was observed for *E. coli* strains bearing single nucleotide substitutions in the 23S rRNA, in particular, A2058G or U2584C, which caused resistance to macrolide antibiotics, or klebsazolicin and some oncocin-like insect PrAMPs, respectively[3,27]. On the other hand, resistance to rumicidins is mediated by the proteins like SbmA involved in their uptake. Here, we also report an example of the inducted mutation that increased the selectivity of MacAB-TolC efflux pump to rumicidins and other PrAMPs. The wild-type and obtained resistant strains showed similar growth rates (Supplementary Fig. 9b), therefore the physiological costs of evolving resistance to rumicidin-1 in a rich medium appear to be minimal. However, the detailed mechanism conferring the mutant MacB[N470D] resistance and its influence on the strain virulence is unclear and requires an additional study. Surprisingly, a strong cross-resistance of this mutant against 'last resort' antibiotic polymyxin B was detected as well (Supplementary Table 7). A 64-fold rise in MIC was shown to be comparable with the effect of MCR-1-mediated colistin resistance reported previously[43]. While this cross-effect is not extended to a key human cathelicidin LL-37 and many other cationic AMPs as well as to different conventional antibiotics (Supplementary Table 7), it should be carefully studied before the potential use of PrAMPs as therapeutics. Together, these results show MacAB-TolC has a broader role in antimicrobial adaptation than previously thought and is not limited to the efflux of polymyxins, bacitracin, and macrolides[44].

In total, the obtained data on biological activity indicate a significant antibacterial selectivity which is an important advantage of PrAMPs over many other membrane-active cationic AMPs. Rumicidins have a reasonable activity against mycobacteria and a range of important Gram-negative bacteria that belong to so-called "ESKAPE" pathogens[45]. Moreover, the efficacy of rumicidins was proved in the mouse lethal septicemia model induced by *E. coli*. We believe that the obtained detailed information about the discovered original peptides interacting with the ribosomal tunnel would provide insights for the rational design of more potent translation inhibitors, in particular, chimeric peptides carrying the structural elements of rumicidins and other known PrAMPs.

## Methods

### Identification of CATHL Genes in Cetartiodactyla WGS Database

The TBLASTN program was used to identify cathelicidin genes in the whole-genome sequencing (WGS, GenBank) database using conservative cathelin-like domain (CLD) fragment FTVKETVCPRTSPQP-PEQCDFKE encoded by the nucleotide sequence located in the second exon of the cattle procathelicidin-3 (Bac7 precursor, GenBank: NP_776426.1) as a query against all Cetartiodactyla WGS projects deposed in NCBI using the values of the default parameters (matrix: BLOSUM62, gap costs: existence 11, extension 1). Then, the obtained hit DNA contigs (± 3000 bp relative to the query) were analyzed to identify exons within the genomic sequence. The putative elastase processing sites in the fourth exon were predicted based on information about known Cetartiodactyla cathelicidins and also using the ExPASy peptide cutter with neutrophil elastase (http://web.expasy.org/peptide_cutter/). Finally, putative mature cathelicidin sequences

were manually (visually) inspected and additionally analyzed by blasting against AMP databases to classify them.

## Bacterial strains

The following strains were utilized: *Escherichia coli* BL21 (DE3) (Novagen), *E. coli* BL21 (DE3) Star (Novagen), *E. coli* DH10B (Invitrogen), *E. coli* ML-35p, *E. coli* ATCC 25922, *Klebsiella pneumoniae* ATCC 700603, *Pseudomonas aeruginosa* ATCC 27853, *Staphylococcus aureus* ATCC 6538 P, *S. aureus* ATCC 29213, *Bacillus subtilis* B-886, *Micrococcus luteus* B-1314, *Mycobacterium phlei* Ac-1291. The *Mycobacterium smegmatis* strain mc(2)155 was kindly provided by Dr. Tatyana L. Azhikina (M.M. Shemyakin & Yu.A. Ovchinnikov Institute of Bioorganic Chemistry, Russia). The strain *E. coli* SQ110 (Δ*rrnA*, Δ*rrnB*, Δ*rrnC*, Δ*rrnD*, Δ*rrnG*, Δ*rrnH*) and its modified variants *E. coli* SQ110DTC (Δ*tolC*), and *E. coli* SQ110LPTD were from[20]. The strain *E. coli* BW25113 and its knockout variants Δ*sbmA*, Δ*mdtM*, Δ*ompF*, Δ*rpoS*, Δ*macA*, Δ*macB* were from the Keio collection[46]. The antibiotic-resistant strains *Enterobacter cloacae* XDR CI 4172, *Acinetobacter baumannii* XDR CI 2675, *Proteus mirabilis* XDR CI 3423, and *E. coli* MDR CI 1057 (*E. coli* 1057, SRCAMB collection № B-10910) were collected and provided by the I.M. Sechenov First Moscow State Medical University hospital. The detailed characteristics of utilized clinical isolates are presented in ref. 22.

## Expression and purification of the antimicrobial peptides

Nucleotide sequences were designed based on *E. coli* codon usage bias data. At the first stage, the recombinant plasmids for the expression of *N*-terminal fragments of rumicidins (Rum-1[1-22], Rum-2[1-22], Rum-3[1-23]), Bac7[1-22], and PR-39[1-22] were constructed. Briefly, the AMP-coding sequences were generated by the annealing of partially self-complementary oligonucleotides (Supplementary Table 8) followed by PCR and cloning in the pET-based vector as described previously[22]. The expression plasmids encoding full-length rumicidins and rumicidin-1 modified analogs were then obtained using ligation independent cloning procedure[47] (Supplementary Table 8). The expression cassette was composed of the T7 promoter, the ribosome binding site, and the sequence encoding the chimeric protein that included His-tag, the *E. coli* thioredoxin A with the M37L substitution (TrxL), methionine residue, and a target peptide. *E. coli* BL21 (DE3) cells were transformed with corresponding plasmids and grown up from an initial OD$_{600}$ of 0.01 for 24 h at 30 °C under stirring in a shaker at a speed of 220 rpm in ZYP-5052 auto-inducing medium based on lysogeny broth (LB) supplemented with 0.2% lactose, 0.05% glucose, 0.5% glycerol, 1 mM MgSO$_4$, 50 mM Na$_2$HPO$_4$, 50 mM KH$_2$PO$_4$, 25 mM (NH$_4$)$_2$SO$_4$, 100 μg/mL of ampicillin, and trace metals according to Studier[18]. The cultured cells were harvested by centrifugation and sonicated in the 100 mM phosphate buffer (pH 7.8) containing 20 mM imidazole and 6 M guanidine hydrochloride. The clarified lysate was loaded on a column packed with Ni Sepharose (GE Healthcare). The recombinant protein was eluted with the buffer containing 0.5 M imidazole. The eluate was acidified (to pH 1.0) by the concentrated hydrochloric acid, and the fusion protein was cleaved by a 100-fold molar excess of CNBr over the number of methionine residues at 25 °C for 18 h in the dark. The lyophilized products of the cleavage reaction were dissolved in water and loaded on a semi-preparative Reprosil-pur C$_{18}$-AQ column (10 × 250 mm$^2$, 5-μm particle size, Dr. Maisch GmbH). RP-HPLC was performed with a linear gradient of acetonitrile in water containing 0.1% TFA. The peaks were collected and analyzed by MALDI-TOF MS using Reflex III mass-spectrometer (Bruker Daltonics). Purity of the obtained recombinant peptides was monitored using Tricine-SDS-PAGE in 16.5% gel containing 6 M urea[48]. The obtained fractions with corresponding molecular masses (Supplementary Table 2) were dried *in vacuo* and dissolved in water. Cathelicidin LL-37, melittin, and VicBac (of >98% purity for all the peptides) were synthesized using a standard solid-phase method and were kindly provided by Dr. Maxim N. Zhmak and Dr. Sergey V. Sychev.

## Antimicrobial assay

Minimum inhibitory concentrations (MIC) were determined by broth microdilution assay based on Clinical and Laboratory Standards Institute (CLSI) guidelines using BSA in the growth medium to minimize AMP nonspecific adsorption[49]. Bacterial test cultures were grown in the Mueller-Hinton broth (MHB, Sigma) at 37 °C to mid-log phase and then diluted with the 2× MH medium supplemented with 1.8% NaCl (or without salt) so that to reach a final cell concentration of 10$^6$ CFU/mL. 50 μL of the obtained bacterial suspension were added to aliquots of 50 μL of the peptide solutions serially diluted with sterilized 0.1% bovine serum albumin (BSA) in 96-well flat-bottom polystyrene microplates (Eppendorf #0030730011). The optical density (OD$_{570}$) of each well was measured, and the lowest concentration of the tested compound that did not result in bacterial growth after incubation for 24 h at 37 °C and 950 rpm was determined as MIC. To verify MIC values, the respiratory activity of the bacteria was determined. Briefly, 5 μL of 0.5 mg/mL resazurin (Sigma) was added to the wells after 24 h (or after 48 h for *M. smegmatis*) of incubation, and the plate was incubated for additional 3 h. In some cases, MICs were determined in the Lysogeny Broth (LB) or Middlebrook 7H9 medium. The reduction of resazurin to resorufin was recorded. In most cases, no significant divergence of MIC values was observed (within ±1 dilution step). The results were expressed as the median values determined based on at least three independent experiments performed in triplicate.

## Selection of resistant bacterial strains

Resistance induction experiments were performed using the previously described method[22]. This approach allows for monitoring MIC values after each transfer. Briefly, on the first day, the overnight culture of the wild-type bacteria was diluted with the 2× MHB supplemented with 1.8% NaCl (or without salt) to reach a final cell concentration of 10$^6$ CFU/mL. 50 μL of the obtained bacterial suspension were added to aliquots of 50 μL of the peptide solutions serially diluted with the sterilized 0.1% BSA in 96-well flat-bottom polystyrene microplates. After incubation for 20 ± 2 h at 37 °C and 950 rpm, MICs were determined as described above. For each subsequent daily transfer, 2–4 μL of the inoculum taken from the first well containing a sub-inhibitory drug concentration was diluted with 2 mL of the fresh 2× MHB supplemented with 1.8% NaCl (or without salt). Then, 50 μL of this suspension was sub-cultured into the next passage wells containing 50 μL aliquots of the peptide at concentrations from 0.25× to 8–16× of the current MIC of each agent. Multiple repeated passages in the presence of antimicrobial agents were performed for each bacterial strain during the experiment. Bacteria that grew at the highest concentrations of AMPs on the final day were passaged further 3 times on drug-free agar plates before determining the final MIC value. Control serial passages in the absence of the agent were also performed, and the resulting cultures showed unchanged MICs against antibacterial agents.

## Whole-genome sequencing of bacteria

To identify potential mechanisms conferring resistance to rumicidin-1, we performed whole-genome sequencing of the corresponding strains followed by genomic DNA de novo assembly and variant calling. The genome of wild-type *E. coli* 1057 strain (SRCAMB collection № B-10910) was used as a reference[28]. 2 × 100 bp pair-end sequencing of the prepared genomic DNA was performed with an Illumina NextSeq550 and MiSeq platforms (Illumina). Evaluation of read quality was performed using the FastQC software (v0.11.9)[50], then reads were filtered, and adapters were cut with TrimmomaticPE (v0.39)[51]. The SPAdes software (v3.13.0) was used to assemble genomes utilizing both filtered paired-end and unpaired reads[52]. Assembly quality was then evaluated with the QUAST program (v5.0.2)[53]. Gene prediction and annotation of assembled contigs were made with the Prokka program (v1.14.6)[54]. Alignment of paired-end reads on reference genome was made using the BWA-MEM (v0.7.17-r1188) algorithm[55]. To call actual

variants, VarScan software (v2.4.0) was launched with a minimal reported variant frequency set to 0.9[56]. To prove the obtained whole-genome sequencing data, the analyzed genes (*sbmA*, *rpoS*, *macB*) were amplified by PCR using specific primers (Supplementary Table 8) and inserted into the pAL-2T vector (Evrogen). The ligation products were transformed into the chemically competent *E. coli* DH10B cells. The obtained plasmids were sequenced on both strands using the ABI PRISM 3100-Avant automatic sequencer (Applied Biosystems).

## Construction of complementation plasmids

Three different plasmid vectors (pUC-based with the strong constitutive artificial promoter J23119[31], the pQE30-based vector with an IPTG/lactose-inducible hybrid T5lac promoter, and pBAD/myc-HisA with an arabinose-inducible promoter) were used for the preparation of the complementation plasmids overexpressing MacB or MacB[N470D] in *E. coli* BW25113 Δ*rpoS*. Two plasmids were obtained by ligase-independent cloning procedure[47] using designed primer pairs (Supplementary Table 8). Briefly, DNA parts were produced by PCR-amplification of vectors and target *macB* gene from the wild-type *E. coli* 1057 or rumicidin-1-resistant strain (R2) expressing the mutant MacB variant. The DNA fragments purified by gel electrophoresis and having 22–23 bp overhangs were mixed with molar ratios of 2:1 (insert to linearized vector) and were subsequently transformed into chemically competent *E. coli* DH10B cells. pBAD-based complementation vectors were obtained by cloning the target genes with restriction enzymes NcoI and EcoRI followed by ligation and transformation into *E. coli* DH10B. Target plasmids (Supplementary Fig. 12) were isolated from individual clones and then analyzed by DNA sequencing.

## Cell-free protein expression assay

To investigate the effects of AMPs and antibiotics on the coupled transcription/translation process, the test compounds were added to a cell-free protein synthesis (CFPS) reaction mix with a plasmid encoding enhanced green fluorescent protein (EGFP) variant (F64L, S65T, Q80R, F99S, M153T, and V163A) under control of the T7 promoter. The cell lysate was prepared using the *E. coli* BL21 Star (DE3) overexpressed T7 RNA polymerase as described previously[22]. The reaction mix consisted of the following components: 1.2 mM ATP, 0.8 mM UTP, 0.8 mM GTP, 0.8 mM CTP, 2 mM of each of 20 proteinogenic amino acids, 1.5 mM spermidine, 1 mM putrescine dihydrochloride, 0.06647 mM calcium folinate, 170 ng/ml tRNA from the *E. coli* MRE 600 strain, 0.33 mM NAD, 120 mM HEPES-KOH (pH 8.0), 10 mM ammonium glutamate, 175 mM potassium glutamate, 60 mM glucose, 15 mM magnesium glutamate, 2% PEG 8000, 25% *E. coli* BL21 Star (DE3) cell lysate, 1 ng/ml plasmid DNA. The reaction volume was 50 µL. The peptides were dissolved in PBS with the addition of 0.1% BSA. Erythromycin was used in the positive control reactions. The fluorescence of the sample without inhibitor was set as the 100% value. The reaction proceeded for 2 h in 96-well v-bottom black polystyrene microplates (Eppendorf #0030601904) in a plate shaker (30 °C, 1000 rpm). The fluorescence of the synthesized EGFP was measured with the microplate reader AF2200 ($\lambda_{Ex}$ = 488 nm, $\lambda_{Em}$ = 510 nm; Eppendorf). The experimental data were obtained from at least two independent experiments performed in triplicate. Non-linear regression curves were generated using GraphPad Prism v.8.0.1 (GraphPad Software Inc.).

## In vitro translation inhibition assay

In vitro translation reactions were carried out using a system of *E. coli* S30 extracts for linear templates (Promega) in 5 µL with the addition of 100 ng of Fluc mRNA and 0.05 mM of D-luciferin, preincubated for 5 min with 5 µM of the tested compounds. Chemiluminescence was measured every 30 s for 20 min at 37 °C using a VICTOR X5 Multilabel Plate Reader (Perkin Elmer, USA). Two independent experiments were performed, and the curve pattern was the same.

## Detection of the translation inhibitors with pDualrep2 reporter

For in vivo bioactivity test, the reporter strain BW25113 Δ*tolC*-pDual-rep2 was used as previously described[57]. Briefly, 10 µL of solutions of tested peptides (0.2 mM of full-length rumicidin isoforms; 0.5 mM of Rum-1[1-22] and Rum-1[9-29]) were applied to an agar plate contained a lawn of the reporter strain. 5 mg/ml erythromycin (Ery, 1 µL) and 25 µg/ml levofloxacin (Lev, 1 µL) solutions were used as control samples. After being incubated overnight at 37 °C, the plate was scanned by ChemiDoc (Bio-Rad) using "Cy3-blot" settings for RFP and "Cy5-blot" for the Katushka2S protein. At least, two independent experiments were performed, and the observed effects were the same.

## Toe-printing analysis

Toe-print analysis was performed as previously described[20] on two matrices: ErmCL and RST1. Briefly, the PURExpress system (NEB) was used for transcription and subsequent translation, then a reverse transcription reaction was performed. The ErmCL template was generated by PCR with two partially self-complementary oligonucleotides (Erm-F and Erm-R, Supplementary Table 8). GGTTATA ATGAATTTTGCTTATTAAC oligonucleotide was used for reverse transcription. RST1 template was prepared as previously described[58] using PCR primers (T7fwd and NV1). Concentrations of tested compounds in toe-print were 0.5, 5, or 50 µM. 1% (v/v) DMSO water solution was used as a negative control sample. Two independent experiments were performed using both matrices, and the observed effects were the same.

## Peptide bond formation

Initiation complexes and ternary complexes were prepared as described previously[21,59]. Where necessary initiation complexes were supplemented with rumicidin-1 (1 µM) or amicoumacin A (30 µM) followed by additional incubation for 5 min at 37 °C. To analyze formation of dipeptide fMet-Val, 0.2 µM 70S initiation complexes containing fMet-tRNA$^{fMet}$ in the P site were mixed with 0.375 µM ternary complexes EF-Tu·GTP·[$^{14}$C]Val-tRNA$^{Val}$ (444 dpm/pmol) and incubated for 2 min at 37 °C. Then samples were quenched with 1/10 volume of 5 M KOH and hydrolyzed for 30 min at 37 °C. Samples were neutralized with 1/10 volume of glacial acetic acid and analyzed by reversed-phase HPLC on RP8 (Merck) with an acetonitrile gradient in 0.1% trifluoroacetic acid. Percentage of synthesized dipeptide was determined by incorporation of the radioactive label as a ratio of peptide formed to the amount of the 70S ribosomes in the reaction mixture, as described earlier[60]. Experiments were done in three replicates.

## Assessment of bacterial membrane permeabilization

To examine the ability of the peptides to affect the barrier function of outer and inner membranes of Gram-negative bacteria, we slightly modified the previously described procedure[22] with the use of the *E. coli* ML-35p strain constitutively expressing cytoplasmic β-galactosidase but lacking lactose permease, and also containing β-lactamase in the periplasmic space. The state of the *E. coli* ML-35p outer and cytoplasmic membranes was assessed based on their permeability to chromogenic markers nitrocefin (Calbiochem) and *o*-nitrophenyl-β-D-galactopyranoside (ONPG, AppliChem) which are the β-lactamase and β-galactosidase substrates, respectively. The cells were incubated in the trypticase soy broth (TSB) for 16 h at 37 °C, washed three times with phosphate-buffered saline (PBS, pH 7.4) to remove residual growth media, adjusted to the concentration of $2.5 \times 10^8$ CFU/mL, and stored on ice until used. Experiments were performed in 10 mM sodium phosphate buffer (NaPB, pH 7.4) with or without the addition of 0.9% NaCl and/or a mixture of divalent cations (0.5 mM CaCl$_2$ and 0.5 mM MgSO$_4$). The final concentration of *E. coli* ML-35p cells was of $2.5 \times 10^7$ CFU/mL. The concentrations of ONPG and nitrocefin were of 2.5 mM and 20 µM, respectively. Peptide samples were placed in the wells of a 96-well flat-bottom polystyrene

microplate, and optical density of the solution enhanced due to an appearance of the hydrolyzed nitrocefin or ONPG and was measured at 492 and 405 nm, respectively, using a microplate reader AF2200. The final volume in each well was 200 μL. Assays were performed at 37 °C under stirring at 500 rpm. Control experiments were performed under the same conditions without adding a peptide. Two independent experiments were performed, and the curve pattern was the same.

## Hemolysis and cytotoxicity assay

Hemolytic activity of the peptides was tested against the fresh suspension of human red blood cells (hRBC) using the hemoglobin release assay as described previously[61]. Two experiments were performed with the hRBC from blood samples collected from healthy donors by certified medical personnel upon informed written consent. All procedures were approved by the Ethics Committee of the Institute of Experimental Medicine (protocol 1/20 of 2/27/2020) and comply with the ethical principles of the Declaration of Helsinki. The quantitative data were represented as average means with standard deviations. The colorimetric 3-(4,5-dimethylthiazol-2-yl)-2,5-diphenyltetrazolium bromide (MTT) dye reduction assay was used to determine cytotoxicity of the peptides against human keratinocyte cell line (HaCaT) and transformed human embryonic kidney cells (HEK293T). Both cell lines were purchased from American Type Culture Collection (ATCC). $10^4$ cells per well in Dulbecco's modified Eagle's medium (DMEM/F12) supplemented with 10% fetal bovine serum (FBS, Invitrogen) were placed into 96-well plates and then cultured in the $CO_2$-incubator (5% $CO_2$, 37 °C). After the media were removed, the peptides were dissolved in 100 μL of the same medium and added to cell cultures at different final concentrations. 20 h later, 20 μL of MTT (5 mg/mL, Sigma) was added to each well, and the plates were incubated for 4 h at 37 °C. Then, the media were discarded and 100 μL DMSO-isopropanol mixture at a ratio of 1:1 (v/v) was added to each well to dissolve the crystallized formazan. The absorbance at 570 nm was measured by a microplate reader AF2200 (Eppendorf). An optical density in the wells containing cells cultured without the peptides was assumed to represent 100% cell viability. Two independent experiments were performed for each peptide.

## Cytokine response to cathelicidins on human cells in vitro

The acute monocytic leukemia THP-1 cell line (ATCC TIB-202) was cultured in the complete RPMI 1640 medium (Invitrogen) containing 10% FBS, 1× antibiotic-antimycotic solution (Invitrogen), and 0.05 mM β-mercaptoethanol, in the $CO_2$-incubator (5% $CO_2$, 37 °C). THP-1 cells were differentiated into proinflammatory macrophages (MΦ1) according to the previously reported protocol[62]. Primary peripheral blood mononuclear cells (PBMC) collected from a healthy donor were purchased from ATCC (PCS-800-011), thawed, and seeded into a 96-well plate one day prior to the experiment at a density of $2 \times 10^5$ cells/well. Two different cell subpopulations (monocytes and T-/B-/NK-lymphocytes) were isolated from PBMC based on their adherence ability. Macrophages MΦ1 were washed out and seeded into 96-well plates at a density of $10^5$ cells/well one day prior to the experiment. The next day, medium in each well was replaced by a fresh complete RPMI 1640 medium with or without 2 μM of a tested peptide (rumicidin-1 or LL-37). Cell cultures were kept in a $CO_2$-incubator (5% $CO_2$, 37 °C) for 24 h. Culture supernatants were collected 24 h later and stored at −70 °C degrees less than one week prior to the assessment of analytes. 26 analytes were measured at a protein level by multiplex xMAP technology using the MILLIPLEX MAP Human Cytokine/Chemokine Magnetic Bead Immunology Panel kit (HCYTO-MAG-60K-27, Merck): eotaxin-1/CCL11, TGFα, GM-CSF, IFNα2, IFNγ, IL-10, IL-12p40, IL-12p70, IL-15, sCD40L, IL-17A, IL-1RA, IL-1α, IL-9, IL-1β, IL-2, IL-3, IL-4, IL-5, IL-6, IL-7, IL-8/CXCL8, IP-10/CXCL10, RANTES/CCL5, TNFα, TNFβ. Multiplex-based assay read-out was performed using the MAGPIX system (Merck) with the xPONENT 4.2 software (Merck) in accordance with the manufacturer's instruction with overnight incubation of the

samples with primary antibodies. The final analysis was carried out with the MILLIPLEX Analyst v5.1 software (Merck). Measurements were performed twice for each sample. The release of the analytes in control and experimental samples were compared by unpaired two-sample t-test using GraphPad Prism v.8.0.1. The p values ≤ 0.05 were considered significant.

## Systemic septicemia infection mouse model

All animal studies were performed at the State Research Center for Applied Microbiology & Biotechnology (SRCAMB) in Obolensk (Russia), approved by the Institutional Bioethics Committee of the SRCAMB (Animal Research Protocol #VP-2022/1) and performed according to the Russian Federal rules and Directive 2010/63/EU of the European Parliament and of the Council. Experiments were performed with female eight- to ten-week-old BALB/c mice ("Andreevka" laboratory animal nursery FMBA, Russia). The animals were housed in groups of 5 in polycarbonate cages, and the following housing conditions were used: 12/12 h dark/light cycle, ambient temperature of 20 ÷ 22 °C, 50% humidity. Food and water were provided ad libitum. In vivo efficacy of rumicidins was tested in the mouse septicemia model induced by E. coli ATCC 25922. Mice were infected with 0.5 mL of bacterial suspension ($1 \times 10^6$ CFU per animal) in saline supplemented with 2.5% mucin (w/v) via intraperitoneal (IP) injection. A total of 4 groups with 5 animals were used. The first group received ciprofloxacin (as a positive control) administered IP once (1 h post-infection) at a dose of 10 mg/kg. The second group received saline (as a vehicle control) and administered IP once (1 h post-infection). Two other groups received rumicidin-1 or rumicidin-3 each administered IP three times (1, 4, and 8 h post-infection) at a dose of 10 mg/kg or of 5 mg/kg, respectively. Survival was monitored for 7 days. After that survived animals were euthanized by $CO_2$ asphyxiation. The spleen was aseptically removed, homogenized, serially diluted and placed on Endo agar for CFU identification.

## Cryo-EM sample preparation and data collection

Initiation complex (70S/mRNA/fMet-tRNA$^{fMet}$) was prepared as described previously[63,64]. Purified complex (0.3 μM) was incubated with rumicidin-2 (10 μM) and spermine (0.6 mM) at 0 °C for 5 min before application onto EM grids. Quantifoil R1.2/1.3 grids with an additional 2 nm amorphous carbon film were glow-discharged for 10 sec at 15 mA using PELCO easiGlow. A total of 3.0 μl of the sample was applied onto the grids and plunge-frozen in liquid ethane using Vitrobot Mark IV (Thermo Fisher) at 10 °C and 100% humidity. Grids were transferred to Krios Cryo-TEM (Thermo-Fisher Scientific) equipped with a Cs-corrector, and a Falcon II electron detector. Data were collected at nominal magnification of ×75,000 and a pixel size of 0.863 Å/pixel, with a defocus range from 0.5 to 1.8 μm at a total dose of 80 e⁻/Å² per exposure distributed across 32 frames.

## Cryo-EM data processing

Dataset consisting of 3528 movies was imported into Relion v3.1.3[65] for preprocessing. Motion correction was performed using an implementation of the MC2 algorithm[66], followed by defocus and astigmatism estimation in CTFFIND[67] using the sum of PS generated during motion correction. Next, crYOLO 1.7.6[68] was used for particle picking using a re-trained model, resulting in 501000 particles which were extracted at 3.452 Å pixel size which were used for a 3D refinement, followed by 3D classification with the local search of 20°, angular sampling of 1.8° and an offset range/step derived from the corresponding values determined during 3D refinement. 371858 particles were re-extracted with a 1.08 Å pixel size and used for another 3D refinement combined with the local masked refinement of the 50S subunit, resulting in a 2.53 Å cryo-EM map. This map was used for consecutive CTF refinement, including defocus, anisotropic magnification and high-order aberrations followed by another run of 3D

refinement. Next, a 2.25 Å map was used for polishing and extracting of particles with the original pixel size. Finally, particles were imported into cryoSPARC 3.3[69]. 3D classification and Heterogeneous refinement were performed to characterize ribosome conformations. Peptide conformations were found to be identical across different classes. All particles were combined for the focused refinement, aberration values were re-estimated and Local refinement with a mask covering 50S subunit[70] was performed, resulting in the 1.95 Å cryo-EM map. Resolution was estimated in cryoSPARC using default parameters.

## Model building

Both unsharpened and sharpened map (B-factor of −41 Å² was applied) from cryoSPARC were used for modeling of rumicidin-2 in Coot v9.6[71]. For an initial model of the 50S subunit the structure of the *E. coli* 50S ribosomal subunit was used (PDB: 8B0X)[72]. Model was manually curated in Coot, followed by refinement in Coot and Servalcat[73]. Water molecules were added in Coot and validated using the sharpened map which was resampled to a pixel size of 0.3 Å.Visualization and analysis of the structures were performed using Chimera v1.15[74] and ChimeraX v1.3[75].

## Reporting summary

Further information on research design is available in the Nature Portfolio Reporting Summary linked to this article.

## Data availability

Coordinates and charge density map for the *E. coli* 50S ribosomal subunit in complex with rumicidin-2 were deposited in the RCSB Protein Data Bank and EMDB with the accession codes 9D89 and EMD-46632, respectively. Data on bacterial genome sequencing were deposited in NCBI SRA database under accession code PRJNA1168027. The data that support the findings of this study are provided as Supplementary Data 1 and Supplementary Information files. Additional data that support the findings of this study are available on request from the corresponding authors. Source data are provided with this paper.

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

## Acknowledgements

We thank Yury S. Polikanov (University of Illinois at Chicago, USA) for valuable advice regarding some of the experiments as well as for careful reading of the manuscript and critical feedback. This work was supported by the Russian Science Foundation: grant No. 21-74-00100 (identification, expression, purification, and antimicrobial activity testing of rumicidins; analysis of rumicidin-resistant strains); grant No. 22-14-00380 (cytotoxicity and immunomodulatory activity testing); grant No. 21-64-00006 (detection of the translation inhibitors and toe-printing analysis); grant No. 22-14-00278 (cryo-EM and dipeptide formation experiments).

## Author contributions

P.V.P. and T.V.O. contributed to the conception of the work. P.V.P. and R.N.K. performed genome mining, and analyzed genome and transcriptome data; P.V.P. and R.N.K. assembled plasmid constructs; P.V.P., R.N.K., I.A.B., and V.N.S. obtained recombinant peptides; E.B.P., A.P., O.V.Shu, A.G.M. and A.L.K. performed cryo-EM studies and analyzed results; A.P., O.V.Shu, and A.L.K. performed and analyzed results of dipeptide formation experiments; P.V.P., R.N.K., I.A.B. and V.N.S. selected and analyzed resistant strains; P.V.P., R.N.K., I.V.B., I.A.B., V.N.S., S.V.B., V.I.M., I.A.O. and P.V.S. evaluated biological activities of rumicidins and performed biochemical assays; T.I.K., O.V.K., and A.I.B. performed in vivo studies. All authors analyzed data; P.V.P., O.V.Sha, A.L.K., A.A.B., O.A.D. and T.V.O. interpreted the results. P.V.P, E.B.P., A.P., A.L.K., and T.V.O. wrote the manuscript. T.V.O. critically revised the manuscript, approved and prepared it for publication. All authors have read and agreed to the published version of the manuscript.

## Competing interests

The authors declare no competing interests.
