## [Transparent Peer Review file · Nature Communications]

Rumicidins are a family of mammalian host-defense peptides plugging the 70S ribosome exit tunnel

Corresponding Author: Professor Tatiana Ovchinnikova

This manuscript has been previously reviewed at another journal. This document only contains reviewer comments, rebuttal letters for versions considered at Nature Communications.

Version 0:

Reviewer comments:

Reviewer #1

(Remarks to the Author)

The revised version of the manuscript of Panteleev et al has addressed some of the concerns with the experiments and the organization of the data presentation. However, even though the study identifies rumicidins as a new family of class I PrAMPs, it does not reveal a novel mechanism of action or any major breakthroughs of what is already known of these antimicrobial peptides. Therefore, in my view, the work may be suitable to be published in a more specialized journal.

Following are some comments about points of the work that remain to be addressed:

-None of the data presented demonstrate that rumicidins kill bacteria due to their ability to inhibit translation by binding to ribosomes and blocking formation of the first peptide bond. In fact, the data presented in Supp. Fig. 13 suggest that rumicidins may primarily act by causing outer membrane damage. By the way, in Supp. Fig. 13, is rumicidin-1 labeled as PhBac29?

Authors claim that the effect of these peptides on membrane integrity may be due to the presence of aromatic Trp and Phe residues in their structure. However, effect of the rumicidin-1 variant W23A,F24A is not shown.

- Lanes 210-213: authors mention that density for the eleven N-terminal residues of rumicidin-2 was not visible, presumably because of the high-mobility of this segment. Therefore, it is extremely confusing that later on, in lanes 228-232, interactions of Tyr7 and Arg10 of rumicidin-2 with tunnel elements are described. Not only there is no figure showing these interactions but, as written in Fig. 2A, Tyr residues are not present in any of the rumicidins and residue 10 of rumicidin-2 is a Pro not an Arg. Authors must clarify.

-Authors show that rumicidin mutants lacking 3-8 N-terminal amino acids preserve their ability to inhibit in vitro translation. Given that such mutants are missing the segment that is claimed to invade the ribosomal A site, would they retain the same mechanism of action of blocking formation of the first peptide bond (a toeprint experiment could clarify this)? This is potentially interesting since if this is the case, one would expect that the binding mode of these shortened peptides inside the exit tunnel should be different compared to that of the wild-type rumicidins.

-Lanes 153-156: conclusions on the stability of the initiation complexes formed with different class I PrAMPs can be drawn only if toeprint experiments at different concentrations of rumicidin-1 and Bac7[1-35] (and/or Tur1A) are done side by side.

-Similar to the above point, SAR differences between rumicidin-1 and Bac7[1-22] can only be revealed by testing side by side the effects of the corresponding variants on translation inhibition and MICs.

- Nomenclature of the E. coli SQ110 -lptD strain is incorrect since LptD is an essential protein. Maybe the mentioned strain carries the imp4213 mutation in the lptD gene which increases OM permeability.

- Lanes 219-227 add reference to Fig. 4f

Reviewer #2

(Remarks to the Author)

Panteleev et al present a thorough study, supported by several lines of experimental evidence, that reveals a family of antibacterial peptides encoded in ruminant genomes. This was a strong manuscript to start with; my comments have been fully addressed by the authors, yielding an improved manuscript. I recommend this work for publication.

Version 1:

Reviewer comments:

Reviewer #1

(Remarks to the Author)

In this newly revised version of the manuscript, the authors have improved the presentation of the data and clarify several of the formerly unclear results and conclusions. However, the mechanism of action of rumicidins that the authors have elucidated is not sufficiently different from that reported for other class I Proline-rich peptides. Therefore, from my point of view, the impact of this work to the field is somehow weak.

Reviewer #1 (Remarks to the Author):

The revised version of the manuscript of Panteleev et al has addressed some of the concerns with the experiments and the organization of the data presentation. However, even though the study identifies rumicidins as a new family of class I PrAMPs, it does not reveal a novel mechanism of action or any major breakthroughs of what is already known of these antimicrobial peptides. Therefore, in my view, the work may be suitable to be published in a more specialized journal.

The authors thank the Reviewer for the valuable comments and suggestions. For the convenience of the Reviewer, all the changes and additions in the revised manuscript were highlighted in color. A distinctive feature of class I PrAMPs is that they block PTC and inhibit formation of the first peptide bond. Indeed, since a similar basic mechanism of action is also observed for rumicidins, this new unique family was assigned to the class of oncocin-like AMPs (class I). However, the most essential differential feature of rumicidins as a new family within the class I is a fundamentally new structural organization lacking the consensus site of oncocin-like AMPs. An interesting feature of the rumicidins is also the presence of a structurally variable and mobile N-terminal sites which affect the activity spectrum of the natural orthologs, as well as the presence of the conserved Trp23-Phe24 dyad, which increases the efficiency of both translation inhibition and interaction with membranes. This matter is discussed in more detail below. Previously published landmark studies of PrAMPs have been mainly focused on investigation of the molecular mechanism of action of a particular peptide or on improvement of its therapeutic properties. In this paper, for the first time, we carried out a comprehensive study of an entire structural family of novel PrAMPs, from analysis of the structural organization of the corresponding genes/pseudogenes and study of interactions of rumicidins with their molecular targets to investigation of their therapeutic potential as antibiotics. Thus, in our opinion, Nature Communication as the journal with a broad scope appears to be the most suitable solution to publish this work. The authors thank the Reviewer once again for very useful comments and suggestions which contributed to improve the quality of the paper in accordance with the journal high standards.

Following are some comments about points of the work that remain to be addressed:

Point 1. -None of the data presented demonstrate that rumicidins kill bacteria due to their ability to inhibit translation by binding to ribosomes and blocking formation of the first peptide bond. In fact, the data presented in Supp. Fig. 13 suggest that rumicidins may primarily act by causing outer membrane damage. By the way, in Supp. Fig. 13, is rumicidin-1 labeled as PhBac29?

We thank the Reviewer for the valuable consideration. Indeed, PhBac29 was the original name of rumicidin-1, and it was corrected. The doubts expressed by the Reviewer prompted us to carry out a series of additional experiments in order to clarify membranotropic effects of rumicidins and their analogs. In the previous version of experiments with chromogenic substrates, we did not consider the effect of ionic strength on a peptide activity: cells were washed with PBS and added to peptide solutions serially diluted with water, and the final NaCl concentration in the medium was approximately of 0.4%. In additional experiments we used a new scheme. We decided to take into account the influence of ionic strength, and salts were used at concentrations mimicking physiological conditions, in particular 0.9% NaCl and/or divalent cations (0.5 mM Ca²⁺ and 0.5 mM Mg²⁺). The results showed that rumicidin-1 at a concentration of $\geq 1 \mu\text{M}$ (MIC in the salt-free MH medium) was quite effective in damaging the outer membrane

of bacterial cells washed with sodium phosphate buffer, whereas the addition of 0.9% NaCl and/or divalent cations minimized or even abolished the activity (**Supplementary Fig. 14**). The same was true for Bac7[1-22] in most cases (**Supplementary Fig. 14**), except for the maximum concentration (64 μM , $\geq 64\times$ MIC) when acting on cells in the presence of 0.9% NaCl (the panel “NaPB + 0.9% NaCl”), at which significant outer membrane damage was observed. Thus, in the context of the host immune system operating in a complex environment containing NaCl and divalent cations, the most likely scenario is that rumicidin's ability to disrupt the integrity of the outer membrane is suppressed. Antibacterial activity of rumicidin-1 is inhibited in the presence of salt (MIC value of 4, 8, and 16-32 μM in the MH medium with NaCl, divalent cations, and their mixture, respectively), and the electrical double layer around the cell seems to be a key barrier on the way into the cell of highly charged and relatively hydrophilic PrAMP. Perhaps, translocation of rumicidin-1 into the periplasmic space depends on electrostatic interaction with the outer membrane followed by porin-mediated uptake (the gene knockout of *ompF* resulted in a 4-fold increase in MIC, **Fig. 5a**). Even taking into account possible effects of outer membrane damage under specific low ionic strength conditions by rumicidins and Bac7[1-22], it is highly unlikely that they may result in rapid cell death and can be considered as the main mechanism of action. Similar effects have been described for some other antimicrobials. For example, the AMP thanatin, which targets Lpt complex proteins in the periplasm of Gram-negative bacteria, can effectively disrupt the outer membrane by displacing divalent cations stabilizing LPS [DOI: 10.1038/s41467-019-11503-3], but these effects disappear when the ionic strength is significantly increased and $\text{Mg}^{2+}/\text{Ca}^{2+}$ is added [DOI: 10.1126/sciadv.aau2634]. It is known that for ribosome-targeting aminoglycoside antibiotics (hydrophilic and cationic) penetration into the cell is associated with the displacement of divalent cations as well, which is accompanied by an increase in the permeability of the bacterial outer membrane [DOI: 10.1128/AAC.19.5.777]. Some more hydrophobic mammalian PrAMPs can “switch” the mechanism of action to membranolytic [DOI: 10.3390/ijms21197367], but this occurs at concentrations higher than MICs. These peptides target the cytoplasmic membrane and damage it which leads to cell membrane depolarization and subsequent cell death. The effect of rumicidin-1 on the cytoplasmic membrane was quite modest (the fraction of permeabilized cells was less than 20%) even at a concentration of 64 μM , which is 16-fold higher than the MICs against *E. coli* (**Supplementary Fig. 13**). Finally, knockout of the transporter protein SbmA leads to an increase in MICs to $\geq 64 \mu\text{M}$ (**Fig. 5a**), and hence cell membranes are resistant to the action of sufficiently high concentrations of rumicidin-1. Taken together, our data point to the ribosome as the most likely target of rumicidins, which is supported by (i) in vitro translation inhibition data (**Fig. 2c**), (ii) translation inhibition data in living cells (**Fig. 2d**), (iii) toe-printing data (**Fig. 2e, Supplementary Fig. 4**), (iv) experiments on inhibition of peptide bond formation **Supplementary Fig. 5**, and (v) structural studies of the rumicidin complex with the ribosome by Cryo-EM (**Fig.4**).

Point 2. Authors claim that the effect of these peptides on membrane integrity may be due to the presence of aromatic Trp and Phe residues in their structure. However, effect of the rumicidin-1 variant W23A,F24A is not shown.

The authors thank the Reviewer for the valuable suggestion. To fill this gap, we performed a comparative analysis of the ability of rumicidin-1 and its analogue [W23A,F24A] to increase the permeability of *E. coli* membranes under different environmental conditions. In all cases in the concentration range from 1 to 64 μM , the analog showed a significant decrease in activity (**Supplementary Fig. 14**). This effect was clearly demonstrated under conditions of incubation of

cells with AMPs in the buffer solution with a low ionic strength (**Supplementary Fig. 14**, the panel “NaPB”), promoting the interaction of cationic AMPs with bacterial membranes: for example, similar kinetic profiles were observed for the peptide rumicidin-1 at a concentration of 1 μM and for its analog [W23A,F24A] at a concentration of 16 μM . In addition, at a concentration of 64 μM (2x MIC), the analog caused cytoplasmic membrane damage in 3% of cells after 4 h of incubation, whereas the rate of permeabilized cells for rumicidin-1 at 64 μM (16x MIC) was about 17% (**Supplementary Fig. 13**).

Point 3. - Lanes 210-213: authors mention that density for the eleven N-terminal residues of rumicidin-2 was not visible, presumably because of the high-mobility of this segment. Therefore, it is extremely confusing that later on, in lanes 228-232, interactions of Tyr7 and Arg10 of rumicidin-2 with tunnel elements are described. Not only there is no figure showing these interactions but, as written in Fig. 2A, Tyr residues are not present in any of the rumicidins and residue 10 of rumicidin-2 is a Pro not an Arg. Authors must clarify.

In fact, the interactions of the Api137 peptide (insect-derived class II PrAMP) with the ribosomal tunnel were described in lanes 228-232. The corresponding sentences were edited.

Point 4. -Authors show that rumicidin mutants lacking 3-8 N-terminal amino acids preserve their ability to inhibit in vitro translation. Given that such mutants are missing the segment that is claimed to invade the ribosomal A site, would they retain the same mechanism of action of blocking formation of the first peptide bond (a toeprint experiment could clarify this)? This is potentially interesting since if this is the case, one would expect that the binding mode of these shortened peptides inside the exit tunnel should be different compared to that of the wild-type rumicidins.

This is an interesting discussion point. We assume that these analogs maintain the same position in the tunnel. Based on the overlapping structure data (**Fig. 2a** and **Fig. 4d**), all of them, including Rum-1[9-29], should reach the A site and block the PTC. This assumption is supported by the toeprint assay (**Supplementary Fig. 4**) where we see a strong AUG signal for Rum-1[9-29] similar to the full-length rumicidin-1.

Point 5. -Lanes 153-156: conclusions on the stability of the initiation complexes formed with different class I PrAMPs can be drawn only if toeprint experiments at different concentrations of rumicidin-1 and Bac7[1-35] (and/or Tur1A) are done side by side.

We agree with the Reviewer that a side-by-side experiment would be the best solution. However, we do not have these reference peptides at the moment. As the above considerations are not critical for describing the observed phenomena, we decided to remove the corresponding part.

Point 6. -Similar to the above point, SAR differences between rumicidin-1 and Bac7[1-22] can only be revealed by testing side by side the effects of the corresponding variants on translation inhibition and MICs.

We thank the Reviewer for the valuable consideration. The authors believe that a comparison of the results obtained with previously published SAR studies of the Bac7 peptide allows us to better explain the logic of our design of truncated analogs. We also consider it appropriate to compare the “MIC fold change” values for wild-type peptides and their analogs as these are relative ones.

Point 7. - Nomenclature of the E. coli SQ110 Δ -lptD strain is incorrect since LptD is an essential protein. Maybe the mentioned strain carries the imp4213 mutation in the ltpD gene which increases OM permeability.

The authors thank the Reviewer for the valuable suggestion. We agree with the Reviewer. Indeed, the used strain SQ110LPTD bears a similar to imp4213 mutation - the deletion of 23 codons (Asp330-Asp352) which makes the outer membrane of the strain more permeable to antibiotics. The strain SQ110LPTD was first described in the manuscript of Orelle et al. (DOI: 10.1128/AAC.01673-13). The text and Figure 5 were changed accordingly.

Point 8. - Lanes 219-227 Δ add reference to Fig. 4f

The authors agree with the Reviewer. The text was changed accordingly. Thank you very much for investing your time and dealing with our manuscript.

Reviewer #2 (Remarks to the Author):

Panteleev et al present a thorough study, supported by several lines of experimental evidence, that reveals a family of antibacterial peptides encoded in ruminant genomes. This was a strong manuscript to start with; my comments have been fully addressed by the authors, yielding an improved manuscript. I recommend this work for publication.

The authors would like to thank the Reviewer once again for his valuable comments and suggestions, which greatly improved the quality of the manuscript. Thank you very much for investing your time and dealing with our manuscript.